# Reversible chirality inversion of an AuAg$_x$-cysteine coordination polymer by pH change

Bing Ni [1] ✉, Dustin Vivod [2], Jonathan Avaro[3], Haoyuan Qi [4], Dirk Zahn[2], Xun Wang [5] & Helmut Cölfen [1,6] ✉

Responsive chiral systems have attracted considerable attention, given their potential for diverse applications in biology, optoelectronics, photonics, and related fields. Here we show the reversible chirality inversion of an AuAg$_x$-cysteine (AuAg$_x$-cys) coordination polymer (CP) by pH changes. The polymer can be obtained by mixing HAuCl$_4$ and AgNO$_3$ with L-cysteine (or D-cysteine) in appropriate proportions in H$_2$O (or other surfactant solutions). Circular dichroism (CD) spectrum is used to record the strong optical activity of the AuAg$_{0.06}$-L-cys enantiomer (denoted as L0.06), which can be switched to that of the corresponding D0.06 enantiomer by alkalization (final dispersion pH > 13) and can be switched back after neutralization (final dispersion pH <8). Multiple structural changes at different pH values (≈9.6, ≈13) are observed through UV-Vis and CD spectral measurements, as well as other controlled experiments. Exploration of the CP synthesis kinetics suggests that the covalent bond formation is rapid and then the conformation of the CP materials would continuously evolve. The reaction stoichiometry investigation shows that the formation of CP materials with chirality inversion behavior requires the balancing between different coordination and polymerization processes. This study provides insights into the potential of inorganic stereochemistry in developing promising functional materials.

Chirality is a multi-scale and multi-discipline subject that offers a new perspective to modulate the material properties[1–3]. The hierarchical assembly structure of chiral Au-cysteine (Au-cys) nanoplatelets has recently garnered attention among various chiral systems, due to their even higher complexity compared to biological counterparts, which can be ascribed to the chiral Au-cys configurations[4]. The hierarchy begins with the sub-nanometric -Au-S- chains, which lead to the formation of coordination polymers (CPs), and the intertwined inter- or intra-chain -Au-Au- aurophilic interactions (AuroIs) could connect different chains to form sheets with thicknesses of ≈1.2 nm[4–8]. The sheets could further convolute due to the AuroIs between different sheets, resulting in structures with greater complexity. This structure emphasized the importance of chirality on the complexity of novel structures. Thus, it could be further inferred that manipulating or inverting the chirality could provide more complexity or other aspects in material science. However, this is also challenging and requires intricate designs on the structures. In small molecular chirality, there are several strategies to stimulate moderate responsive behavior of the optical activities (measured by the circular dichroism, CD, spectrum), such as redox reactions, coordination reactions, solution type, small molecule additions, light

[1]Physical Chemistry, University of Konstanz, Universitätsstrasse 10, 78457 Konstanz, Germany. [2]Friedrich-Alexander-Universität Erlangen-Nürnberg (FAU), Department of Chemistry and Pharmacy, Chair for Theoretical Chemistry/Computer Chemistry Centre (CCC) Nägelsbachstrasse 25, 91058 Erlangen, Germany. [3]Center for X-ray Analytics, Biomimetic Membranes and Textile, Empa, Swiss Federal Laboratories for Materials Science and Technology, Lerchenfeldstrasse 5, St. Gallen CH-9014, Switzerland. [4]Faculty of Chemistry and Food Chemistry & Center for Advancing Electronics Dresden (cfaed), Technische Universität Dresden, 01062 Dresden, Germany. [5]Key Lab of Organic Optoelectronics and Molecular Engineering, Department of Chemistry, Tsinghua University, Beijing 100084, China. [6]Deceased: Helmut Cölfen. ✉e-mail: bing.ni@uni-konstanz.de; helmut.coelfen@uni-konstanz.de

illumination, etc.,[9–11], in some compounds like overcrowded alkenes[12], diarylethenes[13], spiropyrans[14], etc. On a larger scale, the advancement in nanoscience recently has kindled new fires in chirality[3,15]. Chirality inversion has been observed by modulating the supramolecular assembly of helical superstructures in certain cases[16–23], triggered by factors such as light, temperature, pH, coordination reactions, etc. The building blocks generally involve a combination of chiral units and chromic molecules, and the assembling structures are typically sustained by π-π interactions, coordination, electrostatic forces, and/or hydrogen bond networks[21,24]. Stimuli on the responsive functional groups within the building blocks could induce the tuning of the supramolecular assembling structures, resulting in the chirality responsive behavior. Inorganic responsive chiral systems offer alternative strategies to modulate the chirality, such as modulating optical activity by magnetic fields[25], erasing and re-creating the chiral configurations[26], modifying the twisting assembling structures[27], etc.

Enlightened by the structural features of Au-cys based CPs and the requirement of responsive materials, it might be interesting to explore whether the chiral structure could be responsive. Although the optical properties of chiral Au-cys CP based materials have been widely explored both theoretically and experimentally[4–8,28], their responsive behavior has not been explored. On one hand, the structures of Au-cys CP based materials are complicated and allow vast possible chiral conformations. They are typically amorphous and at least two different structures such as oligomer chains and lamellar structures are proposed[5,8]. The sub-nm sized chiral -Au-S- coordination chains and the non-covalent interactions (AuroIs, hydrogen bonds, salt bridges, etc.) between different chains enable a wide range of conformations. The structure of cysteine allows for more conformational possibilities, as suggested by the protein folding where the ternary structures could be significantly modulated by the cysteine moieties and the disulfide bonds[29,30]. Additionally, the structures could be further tuned by doping. Doping the Au-cys CP materials with other coinage metals effectively modulates their structures, as evidenced by morphological

results and fluorescence emissions[4,8,31,32]. On the other hand, the sub-nanometric features[33] of the chains render most of the functional groups (such as -NH$_2$, -COOH) exposed, which might be sensitive to environments[34]. Tuning interactions between these groups should result in different potential conformations. Altogether, it should be promising to explore whether the Au-cys CP based materials can exhibit responsive behavior under external stimuli, as well as to investigate methods for tuning their properties. The relevant research can contribute to the toolbox of modulating chirality.

Here we confirmed that the chirality of AuAg$_x$-cys CP materials can be almost completely inverted by tuning the pH. When varying the pH, the chirality could be inverted reversibly. For example, the optical activity of the AuAg$_{0.06}$-L-cys CP (denoted as L0.06), as tracked by the circular dichroism (CD) spectrum and the calculated g-factor plot, could be switched to that similar to the D0.06 sample simply by alkalization (pH > 13). The strong optical activities marked by g-factors could be inverted from −0.03 to +0.02. After neutralization (pH < 8), the optical activity could be switched back. Multiple structural changes at different pHs (≈9.6, ≈13), which are associated with the Au-Au(Ag) AuroI structures in the CPs are found by the UV-Vis and CD spectral measurements, respectively. Further controlled experiments on the CP material synthesis kinetics suggested that the covalent bonds and AuroIs would be formed rapidly and then the conformation of the CP materials would undergo continuous adjusting. The reaction stoichiometry hinted that multiple parallel reaction processes would occur simultaneously, which could lead to different CP materials. The stereochemistry of ligands is also important. The preparation of CP materials with chirality inversion behavior requires the balancing between different processes.

## Results

### Chirality inversion behavior

The preparation of these CP materials is simple. They can be synthesized by mixing the proper amount of HAuCl$_4$ and AgNO$_3$ with

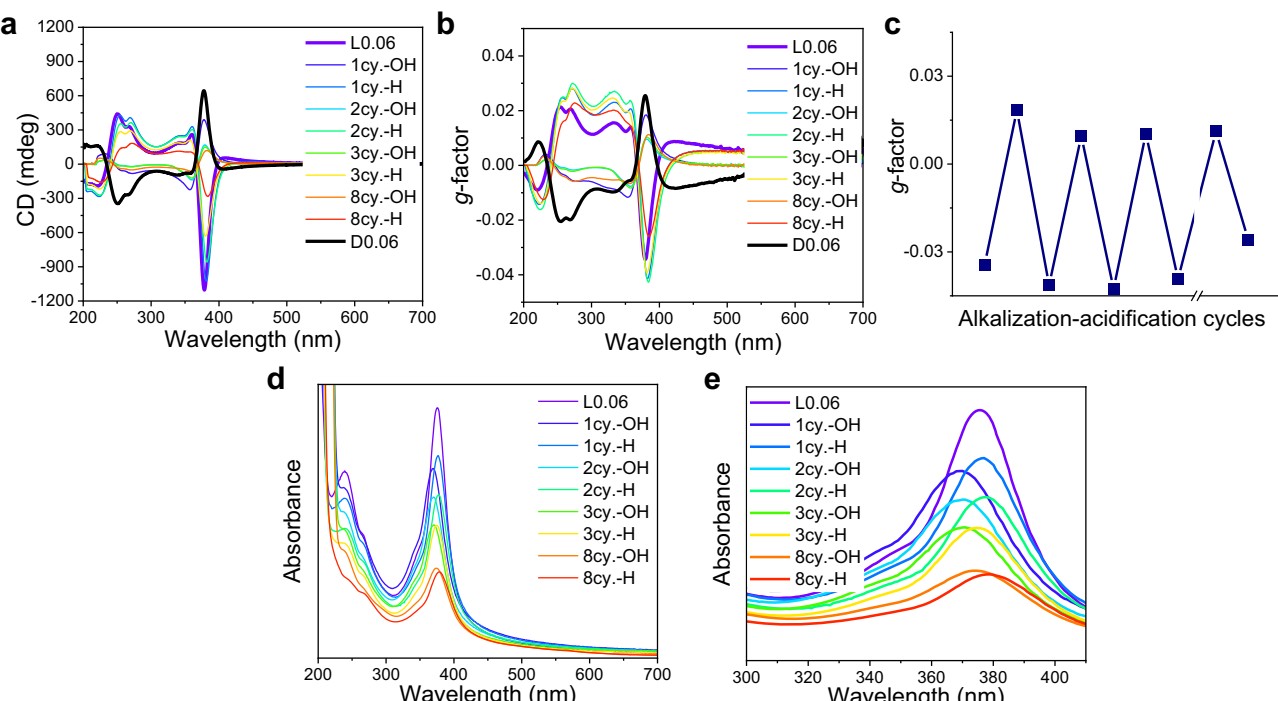

**Fig. 1 | Responsive behavior of L0.06. a** CD spectra of L0.06 upon alkalizations and acidifications (cy. denotes cycles here, -OH denotes the alkalization process while -H denotes the acidification process). The spectrum of D0.06 is added here as a reference (the black line). **b** to eliminate the concentration changes during the

alkalizations and acidifications process, g-factors are calculated. **c** shows the changes of the main g-factor peak of the AuroIs. **d, e** UV-Vis spectra of the L0.06 during the processes. Source data are provided as a Source Data file.

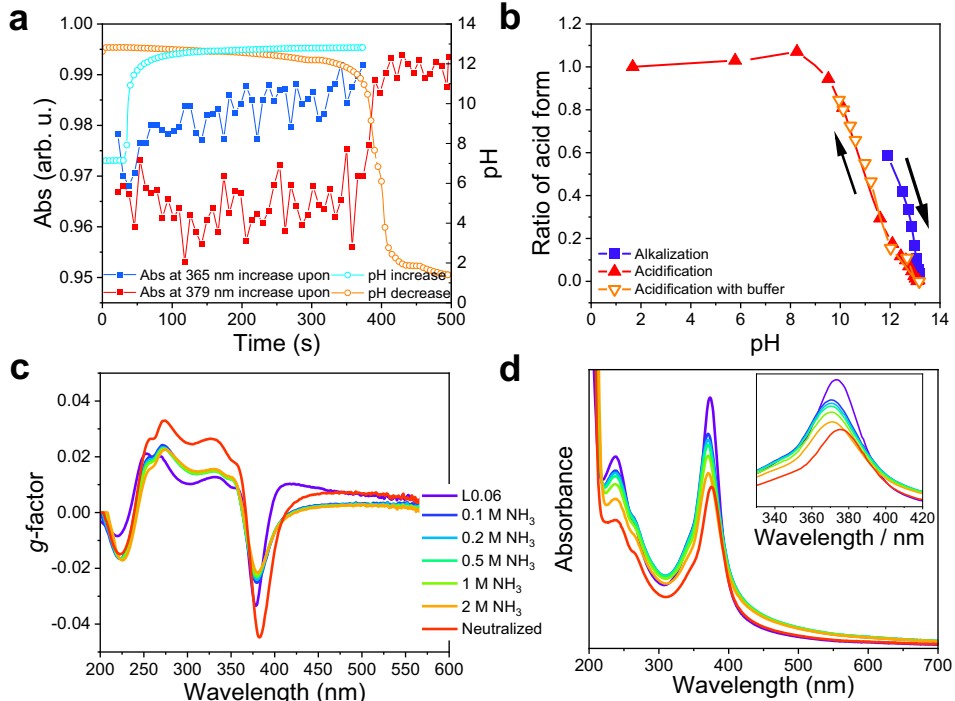

**Fig. 2 | Titration results and responsive behavior in NH₃ solutions. a** UV titration results of L0.06 by checking the absorption at 365 nm and 379 nm during the alkalization and acidification processes, respectively. The whole peak spectra were recorded during the titration and normalized (Suppl. Fig. 7), then the relative absorptions at the corresponding wavelength are extracted to show the trend. **b** CD titration results of L0.06 by recording the CD spectrum at a series of pHs (see details at Suppl. Fig. 8). The change of acid form ratio estimated with and without buffers overlapped well, suggesting that the buffer solution would not interfere with the transitions, and the transition process was gradual. **c**, **d** g-factors and UV-Vis spectra of L0.06 at different concentrations of NH₃. The graphs share the same legend. Source data are provided as a Source Data file.

L-cysteine (or D-cysteine) in H₂O without the addition of any other compounds. White precipitates could be obtained over the course of several hours, the obtained CP materials showed an optical activity with maximal g-factor of −0.04. Such a value is much higher than those of typical small molecules (typically below 0.001, the g-factor of L-cysteine is +0.005 at 215 nm[35])[15,36], suggesting strong optical activity. Nonetheless, the g-factor intensity maximal of such CP materials could be changed from −0.04 to 0.01 during alkalization (Suppl. Fig. 1), suggesting a significant inversion effect. Here we define the measured reversion efficiency (RE) by:

$$RE = \left(1 - \frac{|g_1 + g_2|}{|g_1| + |g_2|}\right) \quad (1)$$

Where $g_1$ and $g_2$ are the g-factor intensity maximal before and after the responsive behavior. If the optical activity is fully inverted, the $g_1 = -g_2$, then the RE = 100%. If the optical activity has the same sign before and after the stimuli, which means the measured optical activity is not flipped, the RE = 0. According to this equation, the RE in this case was 40%. To optimize the inversion property and increase the measured RE, we screened several different surfactants in the synthesis, which typically could influence the polymerization process[37–39]. The responsive behavior could all be attained with the proper doping amount when the growth condition changed (Suppl. Figs. 2, 3). The CP obtained in a cetrimonium bromide (CTAB)-decanol solution showed the most impressive inversion behavior (Fig. 1).

According to Fig. 1a, the CD spectrum of L0.06 was largely inverted during alkalization with the KOH solution (the KOH concentration in the final CP dispersion was 1 M, and the pH was ≈13.7). The intensity of the most intense CD peak could change from ≈ −1100 mdeg to ≈ +400 mdeg. The peaks in the CD spectrum of L0.06 from 300 nm to 450 nm, which originate from the coupling of S-Au bond

localized ligand-metal charge transfer transitions via AuroIs[4,7], showed a symmetric inversion, while the CD spectrum from 215 nm to 300 nm, which arises from the Au-S bonds, was less symmetric (Fig. 1a). Pure cysteine has a peak centered at 215 nm in the CD spectrum, which was not found to invert upon pH changes (Suppl. Fig. 4). The acidification of the pristine L0.06 CP materials showed no significant changes compared to the original states (Suppl. Fig. 5). During the responsive processes, the main peak (378 nm) in the CD spectra would slightly blue-shift during the alkalization and red-shift during acidification, in a range of 15 nm. Such peak shifting behavior was also found in the UV-Vis spectrum (Fig. 1d, e). The as-obtained L0.06 showed a main peak at 378 nm which is a characteristic peak for transitions involving supramolecular Au·cys assemblies[6,8]. The peak would shift to 365 nm and 378 nm during alkalization and acidification, respectively. To exclude the influence of concentration changes during the alkalization and acidification process and get a clearer Fig. about the responsive behavior, g-factors were calculated (Fig. 1b). The intensity of the main g-factor peak could be repetitively inverted from ≈−0.035 to ≈+0.02 several times (Fig. 1c), indicating a measured RE of 73%. The spectra indicate that the optical activity associated with the AuroIs structures (300 nm to 450 nm) could be symmetrically inverted upon pH changes, while the optical activity associated with the Au-S bonds (210 nm to 300 nm) was less symmetrically changed. The same responsive behavior was also found in D0.06 (Suppl. Fig. 6).

To gain more insights into the responsive behavior, we conducted several different titration experiments on the obtained L0.06 CP materials. The UV titration of L0.06 CP materials, by tracking the relative absorbance at a certain wavelength (365 nm in Fig. 2a, full peak spectra in Suppl. Fig. 7a) during the pH titration with KOH titrant, showed that the structural change from the L0.06 original state to the alkalization state was gradual during the pH increasing from 7.1 to 13.2. On the contrary, the change from the alkalization state to the

acidification state (with HCl as the titrant) was sudden (tracking the absorbance at 379 nm in Fig. 2a, full peak spectra in Suppl. Fig. 7b). The switching point of such a process was pH 9.60. The value seemed to be not related to the $pK_a$ of free cysteine molecules (the $pK_a$(-COOH) is 1.96, $pK_a$(-NH$_3^+$) is 8.18, $pK_a$(-SH) is 10.29)[40]. This is reasonable since (1) the -SH group would be deprotonated to form -Au-S-Au- chains after the polymerizations, and therefore change the acid-base properties of the groups; (2) the chemical environments of -COOH and -NH$_2$ groups, before and after polymerizations, might be very different. The CD titration, by measuring the CD spectrum at a series of pHs with KOH or HCl as the titrant, showed that the switching was all gradual (Fig. 2b, Suppl. Fig. 8). During the KOH titration, the pH of the CP material dispersion slowly increased from 12.0 to 13.2, and the CP material gradually evolved to its alkalization states. During the HCl titration, the pH progressively decreased from 13.2 to 8.0, and the polymer gradually changed to its acidification state. The discrepancy during the acidification processes of the UV titration and CD titration indicated that the structure of the L0.06 materials would encounter several different structural changes during pH changes, and the structural changes associated with the UV-Vis spectrum were not necessarily connected with the optical activity changes. To confirm this conclusion, we used a mild alkaline medium (ammonium water) to induce the responsive behavior. When the pH was elevated to ≈12 (corresponding to 1 M NH$_3$), the UV-Vis peak showed a clear blue-shift from 378 nm to 365 nm (Fig. 2d), however, the $g$-factor plots remained almost unchanged (Fig. 2c). Once the solution was further alkalized by KOH (to pH ≈ 13.7), the optical activities would be again flipped (Suppl. Fig. 9). Such results directly support the claim that there exist multiple structural changes during the responsive process. Furthermore, since the hydrogen bond networks are very important to sustain the Au-cys CP materials, and both NH$_3$ molecules and OH$^-$ could in principle change the hydrogen bond networks in the CP materials by forming N-H-X or O-H-X hydrogen bonds, but only one could induce the chirality inversion, thus, it can be inferred that the hydrogen bond networks within the CP materials don't play an important role in the chirality inversion process.

Zeta-potential titrations were further employed to investigate the changes during the chirality inversion behavior (Suppl. Fig. 10). Titration with KOH or NH$_3$ solutions showed similar trends (Suppl. Fig. 10a, b). The KOH zeta-potential titration experiment indicated an isoelectric point of 8.09, suggesting the CP materials are positively charged at pH below 8.09 and negatively charged at pH higher than 8.09. Thus, at the pH switching point (9.60) in the UV-Vis spectra during the transition from alkali state to acidic state, the CP materials are negatively charged. When HCl was used as the titrant, the zeta-potential showed abnormal behavior, which decreased as the pH decreased (Suppl. Fig. 10c). Normally zeta-potential would increase as H$^+$ might bind with the structures as the pH decreases. This could be explained by that the Cl$^-$ coordination in the titrant with the CP materials. Once H$_2$SO$_4$ was used as the titrant, this abnormal behavior disappeared (Suppl. Fig. 10d). KCl solution titration results agreed with this explanation (Suppl. Fig. 10e). Accordingly, the structures contained unsaturated coordination sites where extra ligands such as Cl$^-$ could bind. This might help to explain the discrepancy of NH$_3$ and OH$^-$ in the responsive behavior since their ability to coordinate with the -Au-Au(Ag)- AuroI structures are different[31,32,41]. It would be further interesting to explore whether another kind of ions could induce chirality inversion behavior. Then we explored K$^+$, Cl$^-$, CH$_3$COO$^-$, and SO$_4^{2-}$ as stimuli, and the results showed only the H$^+$ and OH$^-$ could induce such behavior (Suppl. Fig. 11).

## Structural characterization
The amount of Ag$^+$ was found essential to the optical activities and the responsive behavior (Suppl. Figs. 12–14), as well as the nanomorphologies (Suppl. Figs. 15–19). Different amounts of Ag$^+$ could be

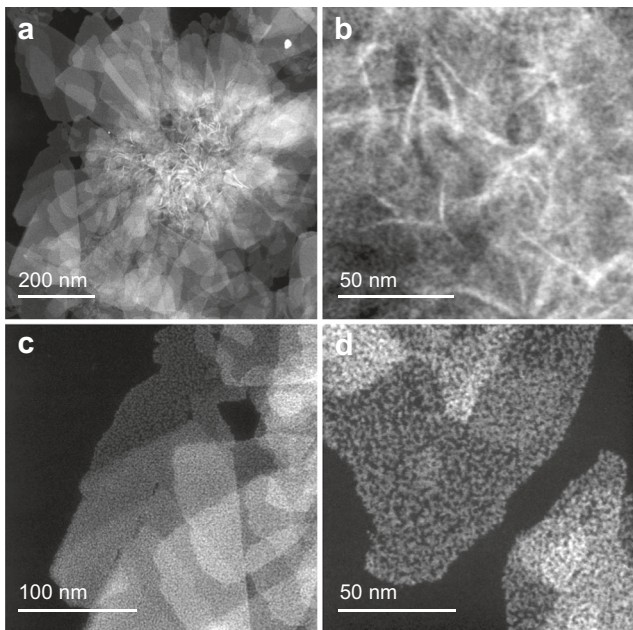

**Fig. 3 | STEM images of the coordination polymers. a–d** STEM images of the L0.06.

doped into the CPs by tuning the precursor feeding ratio (Suppl. Table 1). When Ag$^+$ was excluded in the synthesis, the optical activity of the product (L0) was weak, and the responsive behavior was complicated and not repetitive during each alkalization-acidification cycle (Suppl. Fig. 13). No obvious chirality inversion was found. The scanning transmission electron microscopy (STEM) images of the L0 showed that the morphology was interconnected globular fibers (Suppl. Fig. 15). When Ag$^+$ was introduced into the system, the globular cores would be de-constructed (Fig. 3). The STEM images of L0.06 showed a flower-like morphology, with a ≈ 400 nm core and many flake-shaped petals. Energy dispersive X-ray (EDX) spectrum confirmed the existence of Au, Ag, and S in these petals (Suppl. Fig. 16a). When the doping amount of Ag$^+$ was higher, the morphology would be pure flakes (Suppl. Fig. 17). However, unlike L0.06 CP materials which were stable upon several alkalization-acidification processes, these CP materials with higher Ag contents would be easily destroyed upon alkalization and no chirality inversion behavior can be attained (Suppl. Fig. 14). Thus, we focused on L0.06 for further structural analysis.

The high magnification STEM images (Fig. 3b, d) and selected area electron diffraction (SAED) pattern (Suppl. Fig. 16b) suggested that they were amorphous and composed of randomly interconnected wires. It is worth noting that after screening lots of different flakes of D0.06 samples, a few flakes (2 flakes in screened 60 sheets) showed lattice fringes (Suppl. Fig. 18). Such results on the other hand hinted that the obtained products were a mixture of different structures. The X-ray diffraction (XRD) data showed a series of peaks, which corresponded to layered structures with a spacing of 1.2 nm (Suppl. Fig. 19). Such peaks were consistent with the reported data[4,5,42], and didn't suggest crystalline natures of the CP materials. The X-ray photoelectron spectroscopy (XPS) results indicated that the Ag doping could shift the binding energies of Au and S to lower energies (Suppl. Fig. 20), as Ag features lower electronegativity. Solid-state NMR results indicated that the bonding of H atoms was similar in L0 and L0.06, but the chemical environments of C atoms differed significantly (Suppl. Fig. 21).

Considering that we obtained lattice fringes of 2 flakes of AuAg$_{0.06}$-cys CP, we adopted the parameters as the structural input for the density functional theory (DFT) simulation. Based on the lattice data, we created a series of Au-cys unit cell candidates that would

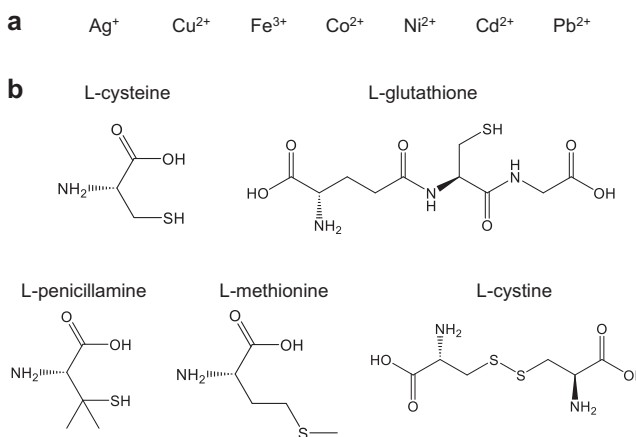

**Fig. 4 | The role of silver ions and cysteine. a** A list of the cations that were tested to replace $Ag^+$ in the formation of L0.06. $Cu^{2+}$ can partially replace the role of $Ag^+$; the rest cannot replace the role of $Ag^+$. **b** A list of the amino acids that were tested to replace cysteine in the formation of L0.06. L-glutathione and L-cystine contain cysteine moiety; L-penicillamine and L-methionine have similar structures as cysteine. But none of them can replace the role of cysteine.

reproduce the observed distances and cell vectors. After eliminating structure models of unrealistic atomic overlap, the proposed structure was refined from DFT calculations (see Methods). The resulting Au-cys unit cell as illustrated in Suppl. Fig. 22 is similar to the reported lamellar crystal structure[4]. However, when the structure was then subjected to 6.125% Au by Ag substitution, strikingly, no convergence was achieved during the relaxation of the 3D-periodic lattice models. We attribute this to the changes in metal-S and metal-N distances, which increase from 2.27 Å and 2.16 Å for Au-S and Au-N to 2.35 Å and 2.20 Å for Ag-S and Ag-N, respectively. It is accepted that the coordination of Ag-cys and Au-cys domains are different and could change the structure[43]. Since the DFT calculations are bound to 3D-periodic unit cell models, the structural insight from such calculations can only be qualitative.

The ideal structure of the Au-cys coordination polymers is -Au-S-covalent chains intertwined by intra- or inter-chain linear -Au-Au-AuroIs (Suppl. Fig. 23)[4–6], which is also observed in the DFT simulations. The formation of these bonds was confirmed by in-situ UV-Vis spectra (Suppl. Fig. 24a, b). When $HAuCl_4$ and $AgNO_3$ precursors were mixed with cysteine, Au-S bonds contributing to linear structures, which have absorptions at around 270 nm[6], appeared immediately. Then the AuroIs help to build the sheet structures, which show peaks centered around 350 nm and 378 nm[6], gradually increased in the first 10 min, and then the shape of the spectra didn't change obviously and only the intensity decreased due to the precipitation of the CP. These suggested that the bond formations (Au-S and AuroIs) were rapid. Nonetheless, the *in-situ g*-factor plots (which ruled out the influence of concentration changes due to the precipitation) obtained via the CD spectroscopy (Suppl. Fig. 24c) suggested that the stereochemistry and the conformation of the CP material continuously underwent changes even though the bonds have been formed, as evidenced by the continuous changes in the plot shapes.

### Controlled experiments to tune the structures

Since the main peaks in the CD spectra and UV-Vis spectra is associated with the AuroIs, the importance of AuroIs were further verified by other metal doping (Fig. 4a). It has been reported that Cu cations[44] could effectively interact with Au(I) as well as Ag cations[45]. Our results suggested that doping the CP with a proper amount of Cu cations could lead to similar chirality switching behavior in the first cycle, but the second cycle only showed a shifted CD spectrum but not chirality inversion anymore (Suppl. Fig. 25). TEM images showed that the Cu cation doping would result in layered sheets and small nanoparticles

(Suppl. Fig. 26). EDX mapping suggested that the Cu could be doped into the structures, while the SAED pattern indicated that the structure is still amorphous (Suppl. Fig. 27). The response with other metal doping like $Fe^{3+}$, $Co^{2+}$, $Ni^{2+}$, $Cd^{2+}$, and $Pb^{2+}$ doping showed similar responsive behavior and structures compared to the L0 (Suppl. Figs. 28, 29) since they cannot strongly interact with Au(I) (though $Pb^{2+}$ could weakly interact with Au(I) under specific conditions[46]).

When cystine, methionine, glutathione, or penicillamine, which features similar chemical structures with cysteine, was used to replace cysteine (Fig. 4b), only penicillamine could form a CP with Au(I). Nonetheless, the $AuAg_x$-penicillamine CP materials didn't show peaks at around 350 nm to 400 nm according to the UV-Vis spectrum, and also no responsive behavior was observed (Suppl. Fig. 30).

We were then interested in the reaction between $HAuCl_4/AgNO_3$ with cysteine. The reaction process between $HAuCl_4$ and cysteine was suggested to be[5,6,28,47]:

$$Au(III) + RSH \rightarrow Au(III)SR + H^+ \tag{2}$$

$$\begin{aligned} Au(III) + 2RSH &\rightarrow Au(I)(SR)_2(\text{intermediated structure}) \\ &+ 2H^+ \rightarrow Au(I) + (SR)_2 + 2H^+ \end{aligned} \tag{3}$$

Without losing generality, we used RSH to denote cysteine here. The first reaction is very fast, while the second reaction is relatively slower due to the electron transfer process[5,28,47]. The product $(SR)_2$ (cystine) has been verified previously[47]. To form the Au-cys CP, the Au(I) formed in reaction (2) further coordinates with RSH (cysteine) with a ratio of 1:1 to form the coordination chains, and the AuroIs could be formed simultaneously or in a step-by-step manner[48]. Thus, the stoichiometry between $HAuCl_4$ and cysteine should be at least 1:3 to form the CP with a high yield if no other reducing agent is added. This stoichiometry has been generally applied in the previous synthesis of Au-CP materials[5–7]. In the preparation of $AuAg_x$-cys CP materials, the ratio was 1:3.2 (which allowed potential errors during the solution handling). The ratio of metal (Au+Ag) to S at the final CP products was found to be around 1:0.98 (L0) 1:1.05 (L0.06), and 1:0.98 (L0) and 1:0.98 (L0.06), according to the XPS and ICP-OES, respectively.

To further explore the relation between the Au(III) reduction reaction and the formation of CP materials, we conducted additional experiments. From this section, the amount of $HAuCl_4$ would be normalized and denoted as 1 part, and the amounts of other reactants would be calculated according to this. If the amount of cysteine was larger than 3 parts, $AuAg_x$-cys CP materials with similar optical properties could be obtained (Suppl. Fig. 31a, b). However, if the amount was just 2 parts, CP materials could be obtained as well (as evidenced by the formation of white precipitation and the absorption band in the UV-Vis spectra). This kind of CP materials has not been well recognized previously. Subjecting them to responsive studies suggested no chirality inversion behavior (Suppl. Fig. 31c, d). Cystine is able to bind with cations to form chiral structures[49], however, Au(0) nanoparticles would be obtained when Au(I) reacts with cystine[47]. This was also the case here if we direct react molecular cystine with Au(I). However, Au(I)$(SR)_2$ was an observed intermediates in the reaction between Au(III) and cysteine. Thus, we speculate that the binding between Au(I) and the $(SR)_2$ formed via reaction is different to that between Au(I) and free molecular cystine. The former intermediate is marked as Au(I)-oxidized cysteine to differentiate with the species formed via reacting free cystine molecules with Au(I). Such Au(I)-oxidized cysteine is responsible for the CP formation at lower feeding cysteine amount, which didn't have chirality inversion behavior. At a higher cysteine feeding, CP materials with chirality inversion behavior could be obtained, which means that the Au(I)-oxidized cysteine could further react with free cysteine to achieve another CP material. From this section, the

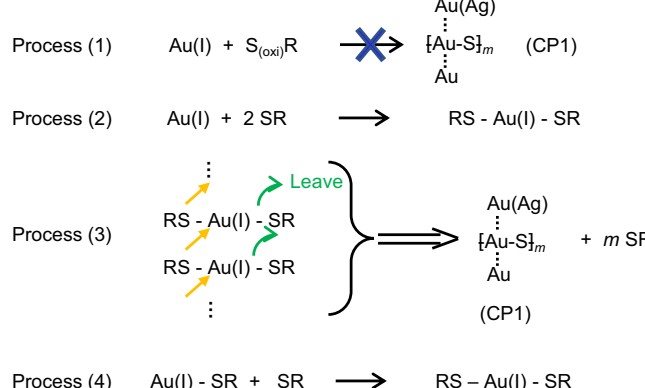

**Fig. 5 | Different processes in the growth of CP materials.** This scheme mainly emphasizes the stoichiometry, but not the chemical reaction mechanisms. Process (1) marks that a 1:1 ratio of Au(I) to cysteine (SR) or oxidized cysteine (S$_{(oxi)}$R) cannot lead to the formation of CP1 which features the impressive chirality inversion behavior. Process (2) marks that one Au (I) species can react with two SR to form the RS-Au(I)-SR species. Process (3) shows that the S groups in one RS-Au(I)-SR species can react with the Au(I) in another RS-Au(I)-SR species and cause the leaving of excess SR groups to form linear covalent chains. This would eventually lead to the formation of CP1 and the release of excessive SR species. The released SR species can bind with the Au(I)-SR species to form RS-Au(I)-SR species (Process (4)), which can go into Process (3).

obtained CP material with chirality inversion behavior will be called CP1, the rest would be just called CP materials.

The reduction reactions and polymerization processes were convoluted during the process, thus complicating the analysis. To overcome this problem, we separated the reduction and polymerization by adding ascorbic acid (AA) to the system. AA could rapidly reduce the Au(III) to Au(I) in the CTAB-decanol system, but cannot further reduce the Au(I) to Au(0) without additional seed particles[50–52]. Taking advantage of this, our strategy worked in this way: (1) HAuCl$_4$ was added into the CTAB-decanol system first; (2) a portion of cysteine was added to allow the formation of oxidized cysteine (if we need to explore the role of oxidized cysteine), which inevitably reduce a portion of Au(III) species; (3) AA was added to fully reduce the rest Au(III) to Au(I); (4) another portion of cysteine was added to allow the growth of CP. If the cysteine amount before AA is limited, the polymerization process could not occur due to the limited amount of ligand and only the oxidation of cysteine could occur. Once the AA is added into the system to fully reduce the Au(III), no reduction processes would further occur in the following periods and only polymerization is possible.

In the first experiment, we tested the reaction between Au(I) and 1 part of cysteine or oxidized cysteine, to see if CP1 could be obtained. The results suggested that Au(I) can react with 1 part of cysteine, but the CP showed a distinct optical feature with CP1 (Suppl. Fig. 32). The products showed broad peaks at ≈250 nm and ≈325 nm in the UV-Vis spectra, and no intensity absorbance at the wavelength of around 378 nm, which contradicts the previous assumption that Au(I) can react with 1 part of cysteine to form CP with characteristic absorbances at 350–400 nm as indicated in the UV-Vis spectrum. Instead, the obtained CP materials showed no strong AuroIs as well as chirality inversion behavior. The same is true for Au(I) and oxidized cysteine (Suppl. Fig. 33). Therefore, the reaction between Au(I) and cysteine (or oxidized cysteine) in a stoichiometry of 1:1 to form CP1 with chirality inversion behavior (process (1) shown in Fig. 5) would not occur.

When the amount of L-cysteine was increased to 1.5 times of the Au(I), CP materials with similar optical properties could be obtained (Suppl. Fig. 34a, b). But only the main peak (centered at ≈370 nm) in the *g*-factor plot can be inverted while the spectra below 330 nm cannot be inverted, which is different to those of CP1. To obtain CP1,

the amount of L-cysteine has to be at least increased to 2 times of the Au(I) (Suppl. Fig. 35). These suggested that process (2) and process (3) are responsible for the formation of CP1. When the cysteine amount was lower than 2 parts (such as 1.5 part), the released SR moieties from process (3) could further bind with the free Au-SR moieties (process (4)), and then enter process (3) again to form CP materials with similar CD and absorption spectra. However, since the conformations need to be continuously adjusted during the growth to form CP1 (as confirmed by the *in-situ g*-factor plots during the growth, Suppl. Fig. 23c), coupling process (3) with process (4) would certainly change the conformation evolution kinetics. Thus, the materials might not evolve to CP1, but CP materials with other conformations, which could only invert the optical activity at ≈370 nm.

Then we seek to understand whether the oxidized cysteine could be incorporated and play a role in the behavior of the CP materials (Fig. 6). When 1 part of oxidized L-cysteine and 1 part of L-cysteine were included in the growth, the obtained CP materials showed a similar *g*-factor plot shape and UV-Vis spectrum as those of CP1 (Fig. 6a, b). However, only the main peak (centered at ≈370 nm) in the *g*-factor plot can be modulated under pH changes. Thus, process (5) and process (6-1) (Fig. 6i) could occur with those reactants, but not forming CP1. It should be noted that 1 part of oxidized L-cysteine could only bind with half of the Au species since one oxidized cysteine molecule requires two cysteine moieties. Thus, the released S$_{(oxi)}$R in the process (6-1) could go back to process (5) again and couple the two processes together. The ideal process (6-1) would occur at an Au(III): L-cysteine ratio of 1:3 with no AA since 2 parts of L-cysteine would already reduce all the Au(III) to Au(I). This process is actually how we obtained our very initial materials discussed in the afore sections, and the obtained materials indeed showed impressive chirality inversion behavior. To further consolidate the conclusion that oxidized cysteine could play a significant role in the material formation and conformations, we tried to include oxidized D-cysteine into the system. When 1 part of oxidized D-cysteine and 1 part of L-cysteine was included in the growth, the obtained CP materials showed distinct spectra with broad peaks at ≈250 nm and ≈325 nm in the UV-Vis spectra, and no intensity absorbance at the wavelength of around 378 nm. Thus, the process (6-1-2) didn't occur. If the amount of oxidized D-cysteine decreased and the L-cysteine increased, CP materials showed intense absorptions at around 378 nm (Fig. 6e–h), which suggested that process (6-2) would occur. But their *g*-factor plots were quite different and the response behavior under pH changes were distinct. No chirality inversion was observed. Thus, the formed CP materials were totally different with the CP1 materials.

Overall, these results strongly suggested that the oxidized cysteine played an essential role in the formation of CP materials here, and the stereochemistry matching between the oxidized cysteine and the free cysteine was also important. Different processes would occur simultaneously, leading to the formation of a mixture of different CP materials. The preparation of CP materials with impressive chirality inversion behavior requires a delicate balance between different processes. This is also consistent with the results that the structure could encounter several different changes at different pHs (≈9.6 and ≈13, respectively) as evidenced by the UV titration and CD titration results.

## Discussion

In conclusion, we have found the chirality inversion behavior of AuAg$_x$-cys CP materials induced by pH changes. The UV-Vis and CD titration experiments showed that the CP materials could encounter several different changes at different pHs (≈9.6 and ≈13, respectively). The main UV-Vis peak of the Au-Au/Ag AuroIs would gradually blue-shift first, and then the optical activity would be inverted at higher pHs. The claim is further confirmed by the controlled experiments with NH$_3$. The ammonia water can only induce the UV-Vis peak shift but not the optical activity inversion, suggesting that the structural changes

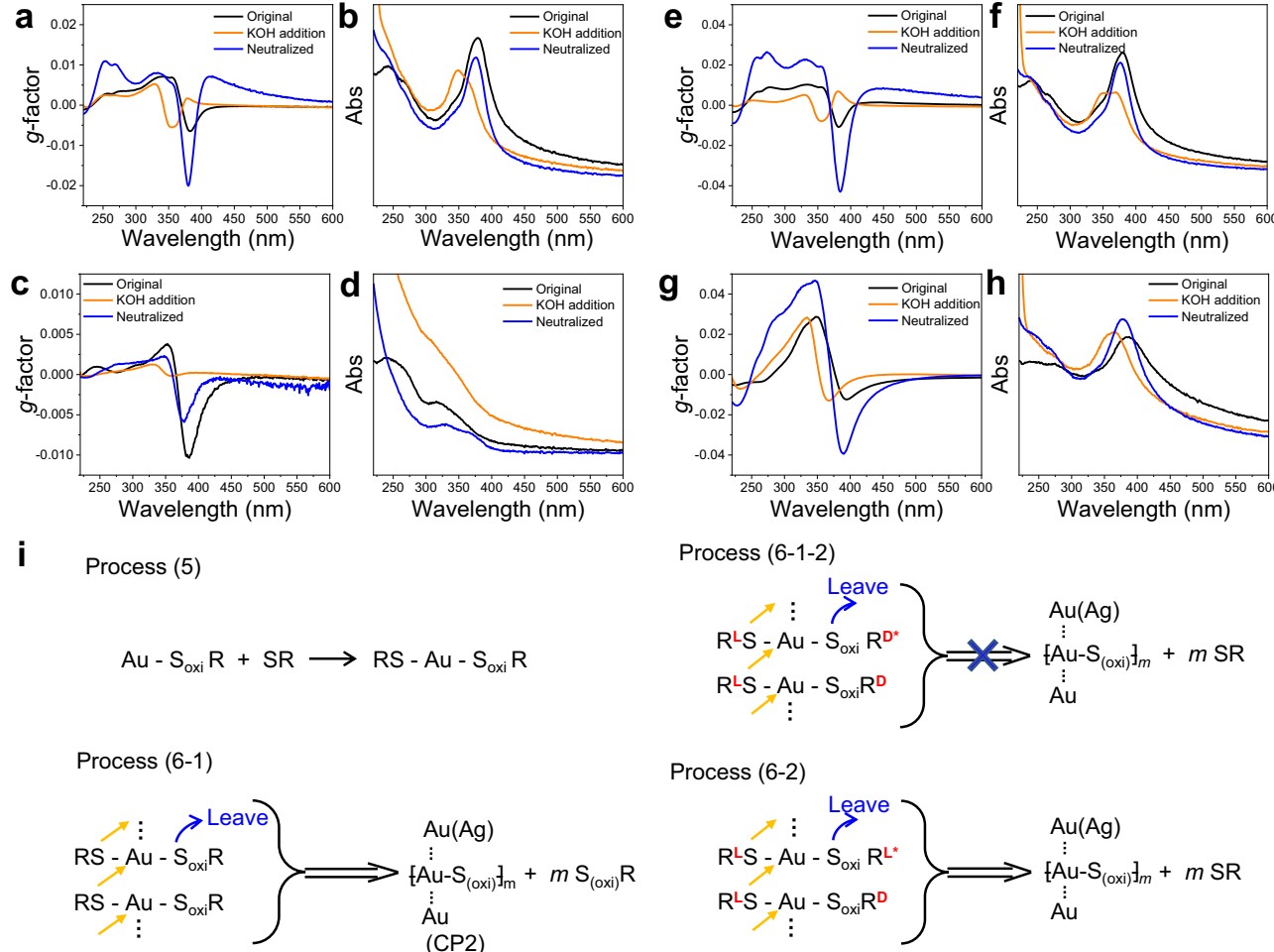

**Fig. 6 | The responsive behavior of CP materials obtained with different amounts of oxidized cysteine and cysteine.** "Original" marks the spectrum of the as-prepared CP materials (black lines), "KOH addition" marks the spectrum after the alkali addition to form a 1 M KOH solution (orange lines), "Neutralized" marks the spectrum after the pH neutralization by HCl addition (blue lines). **a**, **b** CP materials obtained by reacting 1 part of Au(I) with 1 part of oxidized L-cysteine plus 1 part of L-cysteine. The absorption spectrum of the original CP materials shows similar shapes to that of CP1 materials. However, the g-factor plots showed distinct responsive behavior to that of CP1 materials. These suggest that RS-Au-S$_{oxi}$R species can be formed (Process (5) in (i)) and they can further react to form CP materials (called as CP2) (Process (6-1) in (i)). **c**, **d** CP materials obtained by reacting 1 part of Au(I) with 1 part of oxidized D-cysteine plus 1 part of L-cysteine. When

changing the oxidized L-cysteine to oxidized D-cysteine, the obtained materials show no intense absorption peak at around 378 nm. The g-factor responsive plots show no chirality inversion. Thus, the stereochemistry between RS-Au-S$_{oxi}$R species is important to the formation of CP materials, and Process (6-1-2) shown in (i) will not occur. **e**, **f** CP materials obtained by reacting 1 part of Au(I) with 0.5 part of oxidized L-cysteine plus 1.5 part of L-cysteine; The graphs showed similar shapes with that in (**a**, **b**), with intensities different. **g**, **h** CP materials obtained by reacting 1 part of Au(I) with 0.5 part of oxidized D-cysteine plus 1.5 part of L-cysteine. Comparing (**e**–**h**), Process (6-2) shown in (i) can be inferred to form another kind of CP materials different from CP1 and CP2. **i** Summary of the processes concluded from (**a**–**h**). Source data are provided as a Source Data file.

associated with the UV-Vis peak shift are not necessarily related to the structural changes associated with the chirality inversion. Zeta-potential titration experiments further indicated that there exist unsaturated coordination sites in the CP materials, which might be also important to the chirality inversion. The growth kinetics suggested that the -Au-S- covalent bond will be formed rapidly, while the conformation of CP materials would continuously undergo evolution. Further controlled experiments showed the indispensable roles of Au-Au/Ag AuroIs, as well as the importance of cysteine. The reaction stoichiometry hinted that multiple parallel coordination and polymerization reaction processes would occur simultaneously. An excessive feeding of cysteine was necessary to obtain the CP1. Coupling different processes could result in different CP materials, but not all the CP materials could show chirality inversion behavior. The preparation of CP1 materials requires the balancing between different processes. Meanwhile, the oxidized cysteine and their stereochemistry could play a significant role in the process as well. Since the structures

of AuAg$_x$-cys CP materials are too complicated and we were not able to obtain an unambiguous structure, we could not propose a solid chirality inversion mechanism. Nonetheless, the phenomenon of chirality inversion could offer a new perspective on the designing of responsive optical materials.

## Methods

### Chemicals

HAuCl$_4$ • 3H$_2$O (99.99% (metals basis)), L-cysteine hydrochloride (98%), D-cysteine hydrochloride (98%), L-glutathione (98%+), D-methionine (99%), L-methionine (98 + %), L-cystine dihydrochloride (98%), D-cystine (98%), Cu(NO$_3$)$_2$ trihydrate (98%), Fe(NO$_3$)$_3$ nonahydrate (98%), Co(NO$_3$)$_2$ hexahydrate (99%), Ni(NO$_3$)$_2$ hexahydrate (98%), and KOH (99 + %) were purchased from Alfa Aesar. AgNO$_3$ (>99.9%), HCl (37%), and H$_2$SO$_4$ (75%) were purchased from Roth. Hexadecyl-trimethylammonium bromide (CTAB, >99%), ascorbic acid (AA, >99%), and acetic acid (80%), were purchased from Acros

Organics. Hexadecyltrimetryammonium chloride (CTAC, >98.0%), 1-decanol (>98%), L-penicillamine (99%), D-penicillamine (99%), $Cd(NO)_3$ tetrahydrate (99%), $Pb(NO_3)_2$ (99%), and ammonium hydroxide (28%–30%) were purchased from Sigma-Aldrich. Milli Q water was used in all experiments. All reagents were used as received without further purification.

### Synthesis of the AuAg$_x$-cys coordination polymers

In a typical synthesis, 10 μL of $HAuCl_4$ solution (0.05 M), the corresponding amount of $AgNO_3$ solutions (0.01 M), and 16 μL of L-cys solution (0.1 M) were added into a PE tube (2 mL volume) with 2 mL of $H_2O$ or surfactant solutions (CTAC solution, 50 mM; CTAB solution, 50 mM; or CTAB-decanol solution, 50 mM–11 mM). Thus, the concentrations of $HAuCl_4$ and L-cys in the final growth solution were 0.25 mM and 0.8 mM, respectively, yielding a molar ratio of 1:3.2. Then the PE tube was sonicated for around 30 s and placed in a 35 °C water bath without further disturbance. The coordination polymer was collected after 3 h by centrifugation.

### Reduction reaction and polymerization separation

In a typical synthesis, 10 μL of $HAuCl_4$ solution (0.05 M) and 3 μL of $AgNO_3$ solutions (0.01 M) were added into a PE tube (2 mL volume) with 2 mL of CTAB-decanol solution (50 mM–11 mM). To obtain oxidized L-cysteine, a certain amount of L-cysteine solution (0.1 M) (or D-cysteine) was added into the PE tube and sonicated for around 10 s. (5 μL of L-cysteine solution (0.1 M) corresponds to 1 part of oxidized L-cysteine.) Then 15 μL of AA solution (0.1 M) was added into the solution to fully reduce the $HAuCl_4$. After the fully disappearance of the yellow color, another part of L-cysteine was then added. Then the PE tube was sonicated for around 30 s and placed in a 35 °C water bath without further disturbance. The coordination polymer was collected after 3 h by centrifugation.

### Characterizations

A JEOL 2200FS HRTEM operated at 200 kV, was used to obtain the STEM images. A Cary 50 UV-Vis Spectrophotometer from Agilent Technologies and a JASCO J-815 CD spectropolarimeter were used for UV-Vis spectrum and CD spectrum characterizations, respectively. The XRD data were measured with a Bruker D8 with IμS-XR Source. The zeta potential and titration were conducted with a Malvern Zetasizer Nano ZSP equipped with an Autotitrator. FTIR spectra were measured with a Cary 630 FTIR Spectrometer from Agilent. XPS data were measured by ESCALAB Xi+ X-ray Photoelectron Spectrometer from ThermoFisher Scientific. Solid-state NMR spectra were recorded by a JNM-ECZ600R from JEOL.

UV-titration was conducted by coupling the titration experiments with an Ocean optics USB2000 fiber optic spectrometer. Titration experiments were performed using a commercial, computer-controlled system from Metrohm (Filderstadt, Germany), operated with the custom-designed software Tiamo (v2.2). The setup consists of a titration device (Titrando 809) that regulates two dosing units (Dosino 807) capable of dispensing titrant solution in steps as small as 0.2 μL.

CD titration was conducted by manual addition of concentrated KOH solutions into the cuvettes followed by the measurement of the CD spectrum. The pH was recorded by a Mettler Toledo pH meter. All the pH electrodes were calibrated before use.

### DFT calculations

All calculations and relaxations were performed using the PWscf code of the Quantum Espresso software package[53,54]. Atomic cores were represented by norm-conserving Vanderbilt pseudopotentials with a plane wave basis cutoff energy of 50 Ry using the PBE functional with Grimme D3 dispersion[55–57]. Gamma point calculations were used to obtain the structures and energies. Visualization and system setup was done using Vipster (Version 1.19.1b). Initial starting structures were estimated using the TEM data. Optimization of the Au-cys unit cell models was performed in multiple steps. We first kept all heavy atoms fixed whilst refining only the positions of hydrogen atoms. Next the remaining atomic positions were relaxed before finally the entire structure, including the cell edges was optimized to the lowest energy, see also Suppl. Fig. 22. The final Au-cys cell was then explored for Au by Ag substitution. For this, randomly depicted Au atoms were replaced by Ag yielding a AuAg$_{0.06125}$-cys unit cell model, which was relaxed in analogy to the Au-cys models. Strikingly, upon such substitution, no convergence was achieved during relaxation of the 3D-periodic lattice models.

### Reporting summary

Further information on research design is available in the Nature Portfolio Reporting Summary linked to this article.

## Data availability

The data that support the findings of this study are available from the corresponding authors upon request. Source data are provided with this paper.

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

## Acknowledgements

B.N. acknowledges the Alexander von Humboldt Foundation for a PostDoc fellowship. In memory of Prof. Dr. Helmut Cölfen, whose immense contribution to science, education, and mentoring has tremendously shaped our work and academic journey.

## Author contributions

H.C. supervised the project. B.N. found the phenomenon and did the majority of experiments. D.V. and D.Z. conducted the DFT simulations. J.A., H.Q., and X.W. conducted some characterizations and offered suggestions about the project.

## Competing interests

The authors declare no competing interests.
