## [Peer Review File · Nature Communications]

Reversible chirality inversion of an AuAgx-cysteine coordination polymer by pH changeEditorial Note: Parts of this Peer Review File have been redacted as indicated to remove third-party material where no permission to publish could be obtained.

Reviewers' comments:

Reviewer #1 (Remarks to the Author):

The authors report the observation of a very interesting phenomenon, namely the reversible chirality inversion of a silver-doped gold(I)-cysteinide coordination polymer (AuCys) under ambient conditions by simply changing the pH. There is a lot of spectroscopic evidence for this behavior, but the composition and the structure of the samples are not clear and the mechanism proposed for the rearrangements that lead to the changes in optical properties is at best tentative. In fact, there is a lot of confusion regarding both the quality of the samples and the relation of the results to those of previously published work.

The samples are prepared by reacting tetrachloroauric acid trihydrate with L-cysteine hydrochloride in the molar ratio 1:2 (p. 17, not mentioned on p. 5). This makes no sense, because the reagents need to be in the molar ratio 1:3 in order to reach the composition (AuCys), with cystine as the by-product, which needs to be removed in the work-up of the polymer. Two cysteines are not enough to reduce all Au(III) and to complex the resulting Au(I). The 1:3 stoichiometry was used in all previous publications (e. g. ref. 16, 19, 20). The authors cite this work, but give no reason why they chose a different stoichiometry. This point is relevant because the authors claim that their polymer contains disulfide groups -S-S- (below), without offering an explanation in what way these groups would be integrated into the (AuCys) polymer. Silver nitrate is used in small quantities for doping.

The doping with silver salts has also been described previously (ref. 16, 20) using the trifluoroacetate or the nitrate. In this previous work, the samples are described as nanoplatelets, nanoparticles or supramolecular assemblies which show enhanced NL properties or chiroptical effects. The differences (details of preparation, pretreatment of the polymers, spectroscopic and chiral properties etc.) shown by the already documented samples from those now presented in the communication of the authors are not clear.

Selected points: From the beginning (Abstract ff.) it needs to be made clear that only L-cysteine was used in this work. Page 4: "suspectable" ? It is not clear in what way UV-Vis absorption spectra would indicate the presence of metallophilic bonding and give "partial structural information". Are the samples emissive under UV radiation ? P. 5: There is no experimental evidence for the presence of disulfide bonds. Raman spectra could give direct information. P. 9: It is not true that ammonia would "strongly coordinate" to sulfur-bonded silver atoms. P. 10: In situ UV-Vis spectral monitoring the titration of tetrachloroauric acid with cysteine (or the other way round) is difficult to evaluate because cystine and other by-products are also present in the solutions at this stage. The XPS spectra of the products support the (AuCys) 1:1 composition with gold solely in the Au(I) oxidation state as previously determined, but does not indicate the presence of disulfide (ref. 19, 20). P. 12: In the DFT calculations the presence of disulfide bonds is also postulated. The models presented in Fig. 4 show not only Au-S-S-Ag, but also Au-N-Ag linkages. The latter are unlikely, because Au(I) would always prefer Au-S-Ag bridges over Au-N-Ag bridges. The crystal structure of $\text{NH}_4 [\text{Ag}_2(\text{HCys})\text{Ag}(\text{Cys})\cdot\text{H}_2\text{O}]$ (ref. 31) shows, that the coordination sphere of Ag(I) also has a clear preference for S over N. On p. 15, "residual groups near the disulfide bonds" are said to "have a strong effect on the dynamic feature of stereochemistry" referring to ref. 34, but this paper deals with metal-free examples. On the same page, the authors discuss the origin and action of cystine units in the silver-doped samples, but offer no solution for the problem.

In summary, the authors report the observation of an extremely interesting phenomenon which was followed up using a large set of photophysical techniques and theoretical approaches, but the data given for the preparation and analytical characterization of the samples are not satisfactory to explain a "disulfide model". On p. 5 it is stated: "The preparation of these coordination polymers is extremely simple" (with "proper amounts"?), and this should be reconsidered.

Reviewer #2 (Remarks to the Author):

This manuscript reports the reversible chirality inversion of an Ag-doped Au-cysteine coordination polymer (CP) that is triggered by pH changes, while the pristine Au-cysteine CP as well as other amino acids similar to cysteine do not show chirality inversion upon pH changes. The authors track the structural changes related to Au-Au and Au-Ag aurophilic interactions by measuring variable-pH UV-Vis and CD spectra, and conclude that pH changes mainly affect these interactions instead of Au-S coordination bonds. Due to the lack of crystal structure, they have performed PXRD, HRTEM and DFT calculations to reveal the possible structure of the Ag doped Au-cysteine CP. By comparing PXRD and STEM data with reported ones, the authors think that the AuAg_x-cysteine CP should be Au-S coordination chains intertwined by intra- and inter-chain linear Au-Au/Ag aurophilic interactions to form amorphous nanosheet, the HRTEM data based on two flakes suggest that the distance of aurophilic interactions is around 0.281 nm. In addition, they have employed DFT calculations based on the HRTEM data and obtained the optimized Au-cysteine and AuAg_{0.1}-cysteine structures, which reveal that the introduction of Ag⁺ ions induces the formation of disulfide bonds and Au-S-S-Ag chains, and probably the dynamic stereochemistry of the disulfide bonds can be stabilized by aurophilic interactions and corresponds to the chirality inversion.

The results reported in this manuscript are rather interesting. But the Au-cysteine aggregate is very complicated, and to my knowledge this work lacks major evidences to support the conclusions. So, I do not think the manuscript can be accepted in the current form. However, if further experiments can address the questions shown below, this manuscript may be potentially acceptable.

1. The major problem is that no experimental evidence confirms the formation of disulphide bond, which instead is suggested by DFT calculations. However, the formation of disulfide bond is the key point of this work, so this question must be addressed.
2. Since only 2 flakes can be found among 60 samples, the DFT calculations based on the two flakes (HRTEM) may not reveal the real structure of bulky sample. On the other hand, the distance of aurophilic interactions is around 0.281 nm assigned from the TEM analysis, so does the optimized structures confirm this value?
3. To track structural changes and confirm the optimized structures, UV-Vis and CD spectra of Au-S bonds and aurophilic interactions are not enough, the bonds such as Au-N, Au-O are also important. Because HAuCl₄ is used, the amino group of cysteine may be protonated and thus does not coordinate to Au ion.
4. I notice that the authors use HAuCl₄ to synthesize samples, which indicates that redox reaction may take place during synthetic process, so the sample may be a mixture. Is it possible the cysteine -SH group be oxidized to C=S or -S-S- bonds?
5. The optimized structure indicates that the valence of Au is +1. Does the EXAFS spectra exclude the Au³⁺ ion?
6. In the synthesis section, the AgCl precipitate was not removed, which apparently influenced the results obtained from the energy dispersive X-ray spectra.

Reviewer #3 (Remarks to the Author):

Incorporation of foreign metals in nanomaterials enhances several of their properties such as catalytic activity, optical and chiroptical properties; hence, it is of high importance for tuning their properties and broadening the scope of applications. In most of the cases, enhancement in specific properties was observed upon alloying due to the synergistic effect. In particular, silver-doping was used as a strategy to enhance the optical properties of helical gold-cysteine supramolecular assemblies. [see DOI : 10.1039/c9cp00829b] This seminal work published in PCCP, presenting exhaustive characterizations (while I must confess here that high-resolution transmission electron microscopy has allowed the authors to gain insights into the lattice for the first time, partially guiding the use of DFT calculations to support suggested crystal structure models at full atomic detail) and also top-level DFT and TD-DFT calculations to figure out the key ingredients in the enhancement effects, is cited in the present manuscript but never in the manuscript it is mentioned that silver-doping was used before as a strategy to enhance the optical (both nonlinear and chiroptical) properties. Thus the novelty is significantly reduced since PCCP was already dealing with helical silver-doped gold-

cysteine supramolecular assemblies.

Concerning the reversible chirality inversion of an AuAgx-cysteine (AuAgx-cys) coordination polymer by simple pH changes, I must confess that it is a quite appealing strategy. But once again, I am concerned by the novelty of this strategy, since reversible chiral transformation in supramolecular helices was already reported in DOI:10.1038/s41598-018-29152-9 where acidic environments shift the helicity to left-handedness while the alkaline conditions reversed the helical structures to righthandedness. Also Chirality reversal, enhancement and transfer by pH-adjusted surfactant assembly was reported in DOI: 10.1039/D0CC07008D

Although the silver doping strategy and the reversible chiral transformation in supramolecular helices are certainly the most remarkable results of this work, given the pioneering work cited above, I am sceptical about the novelty criterion that justifies publication in Nature Communications.

Again screening several different surfactants in the synthesis which typically could influence the polymerization process and therefore affecting the chiral properties is not new. Indeed, surfactant-induced chirality transfer, amplification and inversion was already reported in cucurbit[8]uril-viologen host-guest supramolecular system (DOI: 10.1039/D1TC03975J).

That said, I encourage the authors to better evaluate the novelty of their work in Introduction based on the reported seminal work on silver doping strategy and reversible chiral transformation in supramolecular helices.

For these reasons, I request that this article be transferred to Communications Chemistry or Scientific reports.

Other comments:

Ref. 18 (DOI: 10.1039/C3SC22215B) does not present any chiroptical measurements neither d= silver-doping results, Thus the sentence "The peaks in the CD spectrum of L0.06 from 300nm to 450nm, which arise from the Au-Au/Ag AuroIs[18], showed a symmetric inversion, while the CD spectrum from 210nm to 300nm[18]" should be better argued. The strong CD band from 300nm to 450nm may likely found its origin in a geometrical structure which couples together the S-Au bond localized LMCT transitions in a chiral way. (see DOI: 10.1016/j.pnsc.2016.08.008).

Fig S5 should report data on pristine coordination polymers (L0 I guess ?). Thus why does the caption mention L0.2 ? In the same vein, does surfactant was added to L0 solutions, results for alkaline conditions ?

The authors should better discuss the similarity/difference observed with spectra reported in Fig. S7 between pristine and silver doped supramolecular assemblies with the CD and UV/vis spectra reported on the seminal paper DOI : 10.1039/c9cp00829b (to better understand the shift observed as a function of doping level in spectra).

Of note, these results indicate that the structures of the obtained AuAgx-cys coordination polymers are Au-S covalent chains intertwined by intra- or inter-chain linear Au-Au/Ag AuroIs to form amorphous nanosheets, and the thickness of the sheets was 1.2nm. This thickness is similar to the thickness reported by Attila Bóta and coworkers (<http://dx.doi.org/10.1016/j.colsurfa.2015.01.048>) on supramolecular chemistry of gold and L-cysteine.

Looking at STEM images of the L0.06, I am thus a bit confused, these images look more similar to "beta-sheet" than "alpha-helix" motifs. Also how realistic are the structural motifs found by DFT calculations as compared to experimental data. Why not trying beta-sheet motifs for calculations ? How do these calculated structures compare with those reported in DOI: 10.1039/c9cp00829b (which nicely explain silver doping effect in optical properties of gold-cysteine supramolecular assemblies). More comments on the relative ratio of alpha vs beta motifs (that would lead to completely different optical properties see DOI: 10.1039/c9cp00829b) should be added.

I agree that according to my knowledge, this is the first time that obtaining the fringes of Au(Ag)-cys coordination polymers is reported. However, the purity is not really controllable and satisfactory (only 2 flakes of the D0.06 sample showed crystalline fringes over more than 60 different coordination polymers).

Doping the coordination polymers with a proper amount of Cu cations (and other metal cations) to tune chirality behavior and possibly chirality switching behavior is interesting and should be better

highlighted since here is the real novelty of doping strategy (even if the effect of metal doping is not at the level of silver doping). For instance EDX and SAED mapping and/or XPS might permit to better evaluating the level of doping using metal ions other than silver.

Reviewer #4 (Remarks to the Author):

As a computational chemist, I will focus on the theoretical aspects of the manuscript, relying on other Reviewers to assess the experimental part.

In this manuscript, the authors report a pH-controlled chirality inversion of an Au-cysteine coordination polymer upon doping with Ag. The work is thorough and includes a screening of different ligands and cations, which highlights the unique features of Ag and cysteine. Although the operating mechanism is not very clear, the phenomenon per se is quite interesting and could trigger new concepts and ideas in the field of inorganic chirality. After addressing my comments below, I would recommend publication in Nat. Commun.

-The computational details are incomplete; the authors must indicate the density functional, the dispersion correction, and the K-points. Please also indicate that cell relaxation was performed (Figure S19d,e).

-Inputs and direct coordinates of the structures must be fully available in the SI for visualization and reproducibility.

-Figure 4. In (a), all cysteines have NH- and COOH groups. In (b), some cysteines present NH₂ and COO- groups. Could the authors comment on these structural changes as well? Was a NH₂/COO- configuration ever evaluated for (a) (i.e., without Ag)?

-How many Au atoms are in the DFT-optimized Au-Cys unit cell? When one Au is substituted by Ag, how much Ag% is present in the unit cell? By looking at Figure 4b, it seems more than the experimental 10%. Have the authors considered using a larger Au-Cys unit cell (i.e., duplicate one direction) to dilute the Ag atom?

-In Figure S20d,e, the authors compare experimental vs. simulated EXAFS of L0.06. For completeness, I would recommend performing the fitting of L0 as well.

-Suggested literature on DFT of monometallic Au/Ag with cysteine, DOI: 10.1039/D0RA06486F

Reviewer #5 (Remarks to the Author):

In this manuscript, the authors report the dynamic switching of CD signals of AuAgx-cysteine assemblies by tuning pH values. The reversible switching of CD signals of supramolecular assemblies is very common, however, it has not been well-demonstrated in Au(I)-thiolate coordination polymers, and unrevealing the underlying mechanism is important. After carefully reading it, I find out there are three major drawbacks of this work: the first is that the authors did not provide sufficient experimental data to test the proposed structural model of AuAgx-cysteine assemblies obtained by DFT simulation, and it is unclear the overall packing structure of the assemblies from the given data; the second is that the authors did not provide reasonable explanations on how the dynamic stereochemistry of disulfide bridge can change with pH values to result in the spectral change, actually, they even did not provide any experimental evidence that disulfide truly exist in their assemblies; the third is that the authors suppose the inversion of CD signals is related to the change of stereochemistry of the disulfide bridge, but they did not explain why Au-cysteine assemblies without dithiol bridges have very similar UV-vis and CD spectra to those of AuAgx-cysteine assemblies with dithiol bridges. Because of the insufficient characterizations of the assembled structure and the lack of understanding of the relationship between the assembled structure and CD signals in this work,

I do not think the work can make a significant contribution in the field of chirality, so I cannot recommend it for publication in current stage.

1. The authors state that "inverting the chirality of a material is of great interest, as it could provide a novel way to utilize chirality in material science, however, it is also extremely challenging." Actually, there are many reports on dynamic inversion of CD signals in supramolecular assemblies (several examples: *Angew. Chem. Int. Ed.* 2016, 55, 2994-3010; *Angew. Chem. Int. Ed.*, 2019, 58, 785-790; *Chem. Sci.*, 2022, 13, 13623-13630), and these related works should be included in the introduction.

2. The authors claim their assemblies are amorphous, then why they can show lattice fringes. Are the assemblies pure or composed of different types of structures?

3. The illustration of the packing structure should be clearer. It's very difficult for readers to understand the structure of the assemblies in Fig. 4. And the authors are suggested to provide experimental data to support such a structural model, such as the existence of disulfide and Au-N bonds in AuAgx-cysteine assemblies.

4. The authors are suggested to use more characterization techniques (such as IR, XPS, solid state NMR, et al.) to elucidate how the assembled structure changes before and after CD signals are inverted.

5. Additional questions: How are the contents of Ag in the assemblies quantified? Why the UV spectra of AuAgx-cysteine assemblies change with pH values? Why NH₃ cannot induce the reversion of CD signals? Will OH⁻ bind with Au⁺ or Ag⁺ at the edge structure at pH value as high as 13?

6. There are also misspelled words, such as "suspectable"; the figure caption for Figure 4h is missing.

Reviewer #1 (Remarks to the Author):

Comment 1-1: The authors report the observation of a very interesting phenomenon, namely the reversible chirality inversion of a silver-doped gold(I)-cysteinide coordination polymer (AuCys) under ambient conditions by simply changing the pH. There is a lot of spectroscopic evidence for this behavior, but the composition and the structure of the samples are not clear and the mechanism proposed for the rearrangements that lead to the changes in optical properties is at best tentative. In fact, there is a lot of confusion regarding both the quality of the samples and the relation of the results to those of previously published work.

Answer 1-1: We are grateful for your valuable comments and the recognition of your interest in our results. We also appreciate your constructive criticism, and it helped us to improve and enhance the quality of our research. Considering your suggestion, we have done additional experimental work to support our findings. The highlighted part in the point-to-point answer also shows the revision in the manuscript.

Comment 1-2: The samples are prepared by reacting tetrachloroauric acid trihydrate with L-cysteine hydrochloride in the molar ratio 1:2 (p. 17, not mentioned on p. 5). This makes no sense, because the reagents need to be in the molar ratio 1:3 in order to reach the composition (AuCys), with cystine as the by-product, which needs to be removed in the work-up of the polymer. Two cysteines are not enough to reduce all Au(III) and to complex the resulting Au(I). The 1:3 stoichiometry was used in all previous publications (e. g. ref. 16, 19, 20). The authors cite this work, but give no reason why they chose a different stoichiometry. This point is relevant because the authors claim that their polymer contains disulfide groups -S-S- (below), without offering an explanation in what way these groups would be integrated into the (AuCys) polymer.

Answer 1-2: The reviewer is correct. The ratio between HAuCl_4 and L-cysteine is crucial for the preparation of these coordination polymers. Previous reports typically employed a ratio of 1:3 (Ref 19, *Prog. Nat. Sci. Mater.* 2016, **26**, 455-460, now ref. 8 in the revised manuscript; Ref 20, *Phys. Chem. Chem. Phys.* 2019, **21**, 12091-12099, now ref. 9 in the revised manuscript), resulting in coordination polymers with an Au:S ratio of around 10:8 (Ref 19), and 13.4:10.8 or 14.5:11.0 (Ref 20) according to the XPS results. Kotov et. al. used a HAuCl_4 : L-cysteine ratio of 1:6.7, resulting in coordination polymers with an Au:S ratio of 10.97:11.10 (Ref 16 *Science*, 2020, **368**, 642-648, now ref. 5 in the revised manuscript). Their detailed material preparation methods are shown in **Answer 1-3**.

In our synthesis, 10 μ L of HAuCl_4 solution (0.05M), the corresponding amount of AgNO_3 solutions (0.01M), and 16 μ L of L-cys solution (0.1M) were added into a PE tube (2mL volume) with 2mL of H_2O or surfactant solutions (CTAC solution, 50mM; CTAB solution, 50mM; or CTAB-decanol solution, 50mM-11mM). Thus, the concentrations of HAuCl_4 and L-cys in the final growth solution were 0.25mM and 0.8mM, respectively, yielding a molar ratio of 1:3.2. The slightly larger amount of L-cysteine allows for potential errors during solution handling. The obtained coordination polymers had an Au:S ratio of 1:0.98 (L0) and 1:1.05 (L0.06) according to the XPS results. The ICP-OES results suggested a metal (Au+Ag) to S ratio of 1:0.98 (L0) and 1: 0.98 (L0.06).

When the molar ratio was 1:2, the resulting products did not exhibit chirality inversion behavior (Figure R1, Figure S31 in the revised manuscript). Chirality inversion behavior could only be observed with a feeding molar ratio higher than 1:3)

Figure R1 (Figure S33) For L0.06, changing the ratio of H₂AuCl₄ to L-cysteine could significantly tune the optical properties.

(a, b) g-factor plots and the absorption spectra;

(c, d) the responsive behavior of the products obtained with a H₂AuCl₄ to L-cysteine ratio of 1:2.

Thank you again for pointing out the issue with stoichiometry. We have found a way to directly obtain Au(I) and have conducted additional controlled experiments on the **stoichiometry**. The results suggest that: (1) multiple coordination processes may occur simultaneously, resulting in the formation of different coordination polymers. As a result, the final products are a mixture of different structures. (2) The Au(I)-oxidized cysteine species could play a significant role in the formation of coordination polymers that exhibit good chirality inversion behavior. (3) Additionally, the stereochemistry of Au(I)-oxidized cysteine and cysteine could also influence the process.

We were then interested in the reaction between H₂AuCl₄/AgNO₃ with cysteine. The reaction process between H₂AuCl₄ and cysteine was suggested to be^{6,7,29,47}:

Without losing generality, we used RSH to denote cysteine here. The first reaction is very fast, while the second reaction is relatively slower due to the electron transfer process^{6,29,47}. The product (SR)₂ (cystine) has been verified previously⁴⁷. To form the Au-cys CP, the Au(I) formed in reaction (2) further coordinates with RSH (cysteine) with a ratio of 1:1 to form the coordination chains, and the Au(I)s could be formed simultaneously or in a step-by-step manner⁴⁸. Thus, the stoichiometry between H₂AuCl₄ and cysteine should be at least 1:3 to form the CP if no other reducing agent is added. This stoichiometry has been generally applied in the previous synthesis of Au-CP materials⁶⁻⁸. In our preparation of AuAg_x-cys CP materials, the ratio was 1:3.2 (which allowed potential errors during the solution handling). The ratio of metal (Au+Ag) to S at the CP products was found to be around 1:0.98 (L0) and 1:1.05 (L0.06), and 1:0.98 (L0) and 1:0.98 (L0.06), according to the XPS and ICP-OES, respectively.

To further explore the relation between the Au(III) reduction reaction and the formation of CP materials, we

conducted additional experiments. From this section, the amount of HAuCl_4 would be normalized and denoted as 1 part, and the amounts of other reactants would be calculated according to this. If the amount of cysteine was larger than 3 parts, $\text{AuAg}_x\text{-cys}$ CP materials with similar optical properties could be obtained (**Figure S31a, b**). However, if the amount was just 2 parts, CP materials could be obtained (as evidenced by the formation of white precipitation and the absorption band in the UV-Vis spectra), but no chirality inversion behavior was found (**Figure S31c, d**). Cysteine is able to bind with cations to form chiral structures, however, Au(0) nanoparticles would be obtained when Au(I) reacts with cysteine⁴⁷. This was also the case here if we direct react cysteine with Au(I). However, Au(I)(SR)₂ was an observed intermediates in the reaction between Au(III) and cysteine. Thus, we speculate that the binding between Au(I) and the (SR)₂ formed via reaction is different to that between Au(I) and free cysteine. The former intermediate is marked as Au(I)-oxidized cysteine to differentiate with the species formed via reacting free cysteine molecules with Au(I). Such Au(I)-oxidized cysteine is responsible for the CP formation at lower feeding cysteine amount, which didn't have chirality inversion behavior. At higher cysteine feeding, CP materials with chirality inversion behavior could be obtained, which means that the Au(I)-oxidized cysteine could further react with free cysteine to achieve another CP material. From this section, the obtained CP material with chirality inversion behavior will be called CP1, the rest would be just called CP materials.

The reduction reactions and polymerization processes were convoluted during the process, thus complicating the analysis. To overcome this problem, we separated the reduction and polymerization by adding ascorbic acid (AA) to the system. AA could rapidly reduce the Au(III) to Au(I) in the CTAB-decanol system, but cannot further reduce the Au(I) to Au(0)⁴⁹⁻⁵¹. Taking advantage of this, our strategy worked in this way: (1) HAuCl_4 was added into the CTAB-decanol system first; (2) a portion of cysteine was added to allow the formation of oxidized cysteine, which inevitably reduce a portion of Au(III) species; (3) AA was added to fully reduce the rest Au(III) to Au(I); (4) another portion of cysteine was added to allow the growth of CP. If the cysteine amount before AA is limited, the polymerization process could not occur due to the limited amount of ligand and only the oxidation of cysteine could occur. Once the AA is added into the system to fully reduce the Au(III), no reduction processes would further occur in the following periods and only polymerization is possible.

In the first experiment, we tested the reaction between Au(I) and 1 part of cysteine or oxidized cysteine, to see if CP1 could be obtained. The results suggested that Au(I) can react with 1 part of cysteine, but the CP showed a distinct optical feature with CP1 (**Figure S32**). The products showed broad peaks at ~250 nm and ~325 nm in the UV-Vis spectra, and no intensity absorbance at the wavelength of around 378nm, which contradicts the previous assumption that Au(I) can react with 1 part of cysteine to form CP with significant AuroIs as indicated in the UV-Vis spectrum. Instead, the obtained CP materials showed no strong AuroIs as well as chirality inversion behavior. The same is true for Au(I) and oxidized cysteine (**Figure S33**). Therefore, reaction between Au(I) and cysteine (or oxidized cysteine) in a stoichiometry of 1:1 to form CP1 with chirality inversion behavior (*process (1)* shown in **Scheme 1**) would not occur.

Scheme 1. Different processes in the growth of CP materials. This scheme mainly emphasizes the stoichiometry, but not the chemical reaction mechanisms.

When the amount of L-cysteine was increased to 1.5 times of the Au(I), CP materials with similar optical properties could be obtained (**Figure S34a, b**). But only the main peak (centered at $\sim 370\text{nm}$) in the g-factor plot can be inverted while the spectra below 330nm cannot be inverted, which is different to those of CP1. To obtain CP1, the amount of L-cysteine has to be at least increased to 2 times of the Au(I) (**Figure S35**). These suggested that *process (2)* and *process (3)* are responsible for the formation of CP1. When the cysteine amount was lower than 2 parts (such as 1.5 part), the released SR moieties from *process (3)* could further bind with the free Au-SR moieties (*process (4)*), and then enter *process (3)* again to form CP materials with similar CD and absorption spectra. However, since the conformations need to be continuously adjusted during the growth to form CP1 (as confirmed by the *in-situ* evolution of CD spectra during the growth, **Figure S23c**), coupling *process (3)* with *process (4)* would certainly change the conformation evolution kinetics. Thus, the materials might not evolve to CP1, but CP materials with other conformations, which could only invert the optical activity at $\sim 370\text{nm}$.

Figure 5 The responsive behavior of CP materials obtained with different amount of oxidized cysteine and cysteine. (a, b) 1 part of oxidized L-cysteine + 1 part of L-cysteine; (c, d) 1 part of oxidized D-cysteine + 1 part of L-cysteine; (e, f) 0.5 part of oxidized L-cysteine + 1.5 part of L-cysteine; (g, h) 0.5 part of oxidized D-cysteine + 1.5 part of L-cysteine; (i) summary of the processes learnt from (a-h).

Then we seek to understand whether the oxidized cysteine could be incorporated and play a role in the behavior of the CP materials (**Figure 5**). When 1 part of oxidized L-cysteine and 1 part of L-cysteine were included in the growth, the obtained CP materials showed a similar g-factor plot shape and UV-Vis spectrum as those of CP1 (**Figure 5a, b**). However, only the main peak (centered at ~370nm) in the g-factor plot can be modulated under pH changes. Thus, *process (5)* and *process (6-1)* (**Figure 5i**) could occur with those reactants, but not forming CP1. It should be noted that 1 part of oxidized L-cysteine could only bind with half of the Au species since one oxidized cysteine contains two cysteine moieties. Thus, the released $S_{(oxi)}R$ in the *process (6-1)* could go back to *process (5)* again and couple the two processes together. The ideal *process (6-1)* would occur at an Au(III): L-cysteine ratio of 1:3 with no AA since 2 parts of L-cysteine would already reduce all the Au(III) to Au(I). This process is actually how we obtained our very initial materials discussed in the afore sections, and the obtained materials indeed showed impressive chirality inversion behavior. To further consolidate the conclusion that oxidized cysteine could play a significant role in the material formation and conformations, we tried to include oxidized D-cysteine into the system. When 1 part of oxidized D-cysteine and 1 part of L-cysteine was included in the growth, the obtained CP materials showed distinct spectra with broad peaks at ~250 nm and ~325 nm in the UV-Vis spectra, and no intensity absorbance at the wavelength of around 378nm. Thus, the *process (6-1-2)* didn't occur. If the amount of oxidized D-cysteine decreased and the L-cysteine increased, CP materials showed intense absorptions at around 378 nm (**Figure 5e-h**), which suggested that *process (6-2)* would occur. But their g-factor plots were quite different and the response behavior under pH changes were distinct. No chirality inversion was observed. Thus, the formed CP materials were totally different with the CP1 materials.

Overall, these results strongly suggested that the oxidized cysteine played an essential role in the formation of CP

materials here, and the stereochemistry matching between the oxidized cysteine and the free cysteine was also important. Different processes would occur simultaneously, leading to the formation of a mixture of different CP materials. The preparation of CP materials with impressive chirality inversion behavior requires a delicate balance between different processes. This is also consistent with the results that the structure could encounter several different changes at different pHs (~9.6 and ~13, respectively) as evidenced by the UV titration and CD titration results.

Comment 1-3: Silver nitrate is used in small quantities for doping. The doping with silver salts has also been described previously (ref. 16, 20) using the trifluoroacetate or the nitrate. In this previous work, the samples are described as nanoplatelets, nanoparticles or supramolecular assemblies, which show enhanced NL properties or chiroptical effects. The differences (details of preparation, pretreatment of the polymers, spectroscopic and chiral properties etc.) shown by the already documented samples from those now presented in the communication of the authors are not clear.

Answer 1-3: Thank you for the comment. Ref. 16 (*Science*, 2020, 368, 642-648, now ref. 5 in the revised manuscript) focuses on the complexity of Au-cys assembling structures, while doping Ag into the structures can result in nanoplatelets. Their material preparation method was as follows: (a) a CTAB (0.27 M)-HAuCl₄ (0.017 M) solution (referred to as Solution A) was first prepared by dispersing the corresponding precursor in H₂O at 70 °C for 12 hours, (b) 0.2 mL of L-cysteine solution (0.57 M) was added to 1 mL of Solution A. Different amounts of AgNO₃ solution or CuCl₂ solution were added to dope the materials. After shaking for 1.5 minutes, the mixture was kept at 70 °C to assemble for 5 hours. Once the reaction was complete, the upper transparent part of the dispersion was removed using a syringe. The bottom precipitates were washed with DI water three times to remove excess CTAB, cysteine, etc., resulting in easy dispersion of the precipitated particles. The optical properties of their materials are shown in Figure R2.

Figure R2. The optical properties of the Au(/Ag/Cu)-D(L)-Cys reported in *Science*, 2020, 368, 642-648. Reproduced with permission from *Science*, 2020, 368, 642-648, Copyright 2020, The American Association for the advancement of Science.

Ref. 20 (*Phys. Chem. Chem. Phys.* 2019, 21, 12091-12099, now ref. 9 in the revised manuscript) focuses on the non-linear optical properties of Au-cys and Ag-doped Au-cys coordination polymers. The so-called α -Au-Cys and β -Au-Cys are prepared differently. Their material preparation method was as follows: 0.38 mmol of cysteine (not

clear whether D- or L- cysteine was used according to the manuscript) was dissolved in 20 mL of water and 0.5 mL of triethylamine. Then, 0.5 mL of HAuCl₄ (0.127 mmol, 50 mg) was quickly added, and the solution was stirred for 30 seconds by inversion. Quickly after, 1 mL of glacial acetic acid was added to induce precipitation of α -Au-Cys NPs, which were centrifuged (5 min/6000 rpm). The supernatant was removed, and α -Au-Cys NPs were redispersed in water (2 mL with 100 mL of TEA) with a vortex time of 5 minutes. Then, 20 mL of methanol was added to precipitate α -Au-Cys NPs with centrifugation (10 min/6000 rpm). α -Au-Cys NPs were also redispersed in 2 mL of water with 10 mL of TEA and precipitated with MeOH (10 mL)/Et₂O (10 mL). After centrifugation (10 min/6000 rpm), the product was dried in air. For all experiments, the resulting powder was dissolved in water containing 0.1% v/v of TEA (pH~11). To produce β -Au-Cys NPs, the solution containing α -Au-Cys NPs was heated at 70 °C for 16 hours. **Ag doping** was achieved in two steps: first, the α -Au-Cys NPs solution was used, and silver trifluoroacetate (1:20 Au/Ag molar ratio) was added to the solution, which was stirred for 24 hours. Then, in a second step, the solution was heated to 70 °C for 60 hours. Afterward, the solution was purified following the same steps as described for α -Au-Cys NPs. The optical properties of their materials are shown in Figure R3.

Figure R3. The optical properties of the α -Au-Cys, β -Au-Cys, and Ag-doped Au-cys materials prepared in *Phys. Chem. Chem. Phys.* 2019, 21, 12091-12099. Reproduced with permission from *Phys. Chem. Chem. Phys.* 2019, 21, 12091-12099, Copyright 2019, Royal Society of Chemistry.

Our sample preparation method is shown in the **Answer 1-2**. After the synthesis, products were centrifuged and washed with H₂O. The synthesis is more similar to those in Ref. 16 (*Science*, 2020, **368**, 642-648, now ref. 21 in the revised manuscript). The optical properties of L0 and L0.06 prepared in CTAB-decanol are shown here. **It can be seen that the main peak positions, peak shapes, and the peak intensity of these products are quite different, suggesting that different sample preparation protocol could lead to very different products, even although all of them are AuAgx-cys coordination polymers.**

Figure R4. The optical properties of L0 and L0.06. The UV-Vis spectra here are close to those in Figure R2b. The

optical activities differed a lot to those in *Science*, 2020, **368**, 642-648 and *Phys. Chem. Chem. Phys.* 2019, 21, 12091-12099.

Comment 1-4: Selected points: From the beginning (Abstract ff.) it needs to be made clear that only L-cysteine was used in this work.

Answer 1-4: Thank you for pointing this out. We have modified the corresponding parts.

Comment 1-5: Page 4: “suspectable” ?

Answer 1-5: Thanks for the reminder. The corresponding part has been revised.

”

Comment 1-6: It is not clear in what way UV-Vis absorption spectra would indicate the presence of metallophilic bonding and give “partial structural information”. Are the samples emissive under UV radiation?

Answer 1-6: We apologize for not making this clear in the original manuscript. The peak around 376 nm in the UV-Vis absorption spectra is indicative of the metallophilic interactions (Au-Au interactions: *Chem. Sci.* 2013, 4, 1852-1857; *Phys. Chem. Chem. Phys.* 2019, 21, 12091-12099; etc.; Ag-Ag interactions: *J. Am. Chem. Soc.* 2010, 132, 8202–8209; *Coordin. Chem. Rev.*, 2021, 432, 213717; etc.; Au-Ag interactions: *Phys. Chem. Chem. Phys.* 2019, 21, 12091-12099; *Science*, 2020, 368, 642-648; etc.). The “partial structural information” was obtained from the HRTEM images, which showed lattice fringes, but not from the UV-Vis absorption spectra. Such info was used for the DFT simulations. The samples indeed are emissive under UV radiation, this has been explored in previous reports.

Comment 1-7: P. 5: There is no experimental evidence for the presence of disulfide bonds. Raman spectra could give direct information.

Answer 1-7: We took the reviewer's suggestion and conducted Raman spectroscopy measurements using a LabRAM HR Evolution Raman spectroscopy instrument from HORIBA Jobin Yvon. The laser wavelength used was 633 nm, and the power of this wavelength was set to 7.7 mW/cm² (100%), which was the lowest power among the other available wavelengths of the machine (514.5 nm, 22.7 mW/cm²; 532 nm, 22.7 mW/cm²). However, we observed that the coordination polymers tended to “burn” even at a power setting of 10%. The “burned” samples exhibited intense peaks from carbon materials that overshadowed the peaks of the coordination polymer. Additionally, the fluorescence properties of the coordination polymer further affected the quality of the Raman spectra. As a result, the obtained data had low quality and could not differentiate the disulfide bonds (see Figure R5).

Figure R5. The recorded Raman spectra of L0 and L0.06, with a 633 nm laser.

In literature, the formation of a disulfide bond would result in a peak at ~504 nm. However, the signal-to-noise of the spectrum is poor, even if the SERS technique is adopted (Appl. Spectrosc., 2004, 58, 570; J. Raman Spectrosc., 2009, 40, 632; etc.) (Figure R6).

[redacted]

Figure R6. Typical Raman spectra of cysteine and cystine (Appl. Spectrosc., 2004, 58, 570).

We also tried to measure the Raman spectrum of the solution sample and the Raman optical activity using ChiralRaman-2X TM. However, even though the sample was very concentrated, only H₂O peaks can be observed.

Figure R7 Raman spectrum of L0.06 aqueous solution. Only the peaks of H₂O can be observed, and no Raman optical activity was obtained.

Due to these results, as well as the XPS results (see detail at **Answer 1-10**), solid-state NMR results (see detail at Figure S21), we agree with the Reviewer that we don't have solid evidence on the disulfide bond mediated chirality inversion mechanism. Thus, we renounced the claim and removed the mechanism discussion in the revised manuscript, but only emphasized the results on the kinetics study and the reaction stoichiometry study. **Exploration of the kinetics suggested that the covalent bond formation was rapid and then the conformation of the CP materials would continuously evolve. The reaction stoichiometry investigation showed that the formation of CP materials with chirality inversion behavior requires the balancing between different coordination and polymerization processes.**

Comment 1-8: P. 9: It is not true that ammonia would “strongly coordinate” to sulfur-bonded silver atoms.

Answer 1-8: Thank you for bringing this to our attention. Since the atomic ratio of metal to S was approximately 1:1 according to the XPS and ICP-OES results, it is likely that the metal (Au or Ag) was not fully coordinated by S. The

zeta-potential titration results indicated the presence of unsaturated coordination sites in the products, which can be coordinated by Cl⁻. Therefore, it is also possible for NH₃ to bind with the metal centers. We agree with the Reviewer that we don't have direct evidence for this. Thus, we have modified the corresponding discussions:

“...To confirm this conclusion, we used a mild alkaline medium (ammonium water) to induce the responsive behavior. When the pH was elevated to ~12 (corresponding to 1 M NH₃), the UV-Vis peak showed a clear blue-shift from 376nm to 365nm(Figure 2c), however, the CD spectra and the g-factor plots remained almost unchanged (Figure 2d, e). Once the solution was further alkalized by KOH (to pH~13.7), the optical activities would be again flipped (Figure S9). Such results are impressive since both NH₃ molecules and OH⁻ could change the hydrogen bond networks in the CP materials by forming N-H-X or O-H-X hydrogen bonds, but only one could induce the chirality inversion, thus, the hydrogen bond networks within the CP materials don't play an important role in chirality inversion.

Zeta-potential titrations were further employed to investigate the changes during the chirality inversion behavior (Figure S10). Titration with KOH or NH₃ solutions showed similar trends (Figure S10a, b). When HCl was used as the titrant, the zeta-potential showed abnormal behavior, which decreased as the pH decreased (Figure S10c). Normally it would increase as the H⁺ might bind with the structures as the pH decreases. This could be explained by the Cl⁻ coordination in the titrant with the CP materials. Once H₂SO₄ was used as the titrant, this abnormal behavior disappeared (Figure S10d). KCl solution titration results agreed with this explanation (Figure S10e). Accordingly, the structures contained unsaturated coordination sites where extra ligands such as Cl⁻ could bind. It would be further interesting to explore whether another kind of ions could induce chirality inversion behavior. Then we explored K⁺, Cl⁻, CH₃COO⁻, SO₄²⁻ as the stimuli, and the results showed only the H⁺ and OH⁻ could induce such behavior (Figure S11).”

Comment 1-9: P. 10: In situ UV-Vis spectral monitoring the titration of tetrachloroauric acid with cysteine (or the other way round) is difficult to evaluate because cystine and other by-products are also present in the solutions at this stage.

Answer 1-9: We apologize for not clarifying this in our original manuscript. The titrations were performed using the L0.06 CP materials dispersed in H₂O, separated from the polymer preparation system. After synthesizing and washing the L0.06 CP materials to remove cystine or other by-products, they were then dispersed in H₂O for further titration measurements.

Comment 1-10: The XPS spectra of the products support the (AuCys) 1:1 composition with gold solely in the Au(I) oxidation state as previously determined, but does not indicate the presence of disulfide (ref. 19, 20).

Answer 1-10: Yes, the XPS spectra should be very useful to determine the valance state of typical elements. However, The reported -SH and disulfide bonds in the literature showed a wide range of binding energies (162eV~165.1eV) and sometimes overlapping. The peak centers of 2p_{1/2} and 2p_{3/2} are quite close (the labeled “2p_{1/2}” peak is not the shakeup signal of the strong 2p_{3/2} peak), which makes the peak differentiation and binding energy identification difficult. Furthermore, the binding energies are sensitive to the degree of protonation (protonation would shift the binding energy to higher energies), which makes the differentiation of disulfide bonds further difficult.

Our XPS spectra look different to those in Ref 17 (now ref 6; Ref 16, 19, 20 don't include the high-resolution S2p scan in the manuscript) (Figure R8), but we cannot assign the difference to the valence or just degree of protonation.

Figure R8. High-resolution S2p scan of our sample (a) and those reported in the literature (b) (Reproduced with permission from *Colloids Surf. A Physicochem. Eng. Asp.* 2015, 470, 8-14, Copyright 2015 Elsevier B. V.). The figure caption of (b) is: XPS spectra showing the peaks of sulfur in different samples: $(\text{AuCys})_n^\beta$ (blue, green); L-cysteine + elemental gold (cyan); $(\text{AuCys})_n^\beta$ + elemental gold (yellow); elemental gold (reaction product of HAuCl_4 and sodium borohydride) (purple).

Comment 1-11: P. 12: In the DFT calculations the presence of disulfide bonds is also postulated. The models presented in Fig. 4 show not only Au-S-S-Ag, but also Au-N-Ag linkages. The latter are unlikely, because Au(I) would always prefer Au-S-Ag bridges over Au-N-Ag bridges. The crystal structure of $\text{NH}_4[\text{Ag}_2(\text{HCys})\text{Ag}(\text{Cys})\cdot\text{H}_2\text{O}]$ (ref. 31) shows, that the coordination sphere of Ag(I) also has a clear preference for S over N.

Answer 1-11: Since our supplemented results didn't proof the existence of disulfide bonds, we re-calculated all the previous structures and revised the corresponding discussions. parameters as the structural input for the density functional theory (DFT) simulation. Based on the lattice data, we created a series of Au-cys unit cell candidates that would reproduce the observed distances and cell vectors. After

eliminating structure models of unrealistic atomic overlap, the proposed structure was refined from DFT calculations (see Methods). The resulting Au-Cys unit cell as illustrated in **Figure S22** is similar to the reported lamellar crystal structure⁵. However, when the structure was then subjected to 6.125% Au → Ag substitution, strikingly, no convergence was achieved during relaxation of the 3D-periodic lattice models. We attribute this to the changes in metal-S and metal-N distances, which increase from 2.27 Å and 2.16 Å for Au-S and Au-N to 2.35 Å and 2.20 Å for Ag-S and Ag-N, respectively. Since our DFT calculations are bound to 3D-periodic unit cell models, the structural insight from such calculations can only be qualitative.”

Although the failure of structural optimization in another way suggests that doping Ag to the Au-cys CPs can drastically tune the structures, we totally removed the previous results on the AuAg_{0.1}-cys structures and **renounced the claim of disulfide bonds**.

Comment 1-12: On p. 15, “residual groups near the disulfide bonds” are said to “have a strong effect on the dynamic feature of stereochemistry” referring to ref. 34, but this paper deals with metal-free examples.

Answer 1-12: The reviewer is correct. Ref 34 (*J. Am. Chem. Soc.*, 2023, doi:10.1021/jacs.3c00586) discusses the stereochemistry of different metal-free compounds with disulfide bonds. Another recently published paper focuses on tuning the chirality of Cd-cystine materials. In this study, the twisting angles, pitch, width, thickness, and length of Cd-cystine bowtie assembling structures could be continuously adjusted by varying the precursor concentrations, pH, ionic strength, and other factors (*Nature*, 2023, 615, 418-424). Although the authors did not attribute this flexibility directly to the disulfide bonds in their paper, considering all these findings together, it is plausible to conclude that disulfide bonds possess unique flexibility in chiral assembling. **We also tried to test whether the Cd-cystine bowtie particles are responsive under pH changes, however, particles would be totally destroyed upon acidification.**

Comment 1-13: On the same page, the authors discuss the origin and action of cystine units in the silver-doped samples, but offer no solution for the problem.

Answer 1-13: We agree with the reviewer that obtaining direct evidence regarding the behavior of cystine units would be highly valuable in understanding the mechanisms behind chirality inversion. However, due to limitations imposed by the nature of the materials and the available techniques (such as Raman, FT-IR, XPS, solid-state NMR spectra, etc.), we were unable to directly obtain such information. Instead, we conducted additional controlled experiments on stoichiometry to gain insights into the formation of coordination polymers exhibiting chirality inversion behavior (refer to **Answer 1-2**). The results provided valuable evidence suggesting that oxidized cysteine plays a significant role in the observed chirality inversion behavior.

Comment 1-14: In summary, the authors report the observation of an extremely interesting phenomenon, which was followed up using a large set of photophysical techniques and theoretical approaches, but the data given for the preparation and analytical characterization of the samples are not satisfactory to explain a “disulfide model”.

Answer 1-14: Thank you once again for your valuable comments and for recognizing the significance of our results. We have made sincere efforts to address the comments provided and improve our understanding of the interesting chirality inversion mechanism. **The additional growth kinetics suggested that the -Au-S- covalent bond will be formed rapidly, while the conformation of CP materials would continuously undergo evolution. The reaction stoichiometry hinted that multiple parallel coordination and polymerization reaction processes would occur simultaneously. An excessive feeding of cysteine was necessary to obtain the CP1 with chirality inversion mechanism. Coupling different processes could result in different CP materials, but not all the CP materials could show chirality inversion behavior. The preparation of CP1 materials requires the balancing between different processes. These results suggested an**

overlooked aspect in the previous reports. Meanwhile, the oxidized cysteine and their stereochemistry could play a significant role in the process as well. Due to the complexity of the CP materials, we could not propose a solid chirality inversion mechanism. Nonetheless, we believe the novel chirality inversion phenomena could offer a new perspective on the designing of responsive optical materials.

Comment 1-15: On p. 5 it is stated: “The preparation of these coordination polymers is extremely simple” (with “proper amounts”?), and this should be reconsidered.

Answer 1-15: The sentence is modified.

Reviewer #2 (Remarks to the Author):

Comment 2-1: This manuscript reports the reversible chirality inversion of an Ag-doped Au-cysteine coordination polymer (CP) that is triggered by pH changes, while the pristine Au-cysteine CP as well as other amino acids similar to cysteine do not show chirality inversion upon pH changes. The authors track the structural changes related to Au-Au and Au-Ag aurophilic interactions by measuring variable-pH UV-Vis and CD spectra, and conclude that pH changes mainly affect these interactions instead of Au-S coordination bonds. Due to the lack of crystal structure, they have performed PXRD, HRTEM and DFT calculations to reveal the possible structure of the Ag doped Au-cysteine CP. By comparing PXRD and STEM data with reported ones, the authors think that the AuAg_x-cysteine CP should be Au-S coordination chains intertwined by intra- and inter-chain linear Au-Au/Ag aurophilic interactions to form amorphous nanosheet, the HRTEM data based on two flakes suggest that the distance of aurophilic interactions is around 0.281 nm. In addition, they have employed DFT calculations based on the HRTEM data and obtained the optimized Au-cysteine and AuAg_{0.1}-cysteine structures, which reveal that the introduction of Ag⁺ ions induces the formation of disulfide bonds and Au-S-S-Ag chains, and probably the dynamic stereochemistry of the disulfide bonds can be stabilized by aurophilic interactions and corresponds to the chirality inversion.

The results reported in this manuscript are rather interesting. But the Au-cysteine aggregate is very complicated, and to my knowledge this work lacks major evidences to support the conclusions. So, I do not think the manuscript can be accepted in the current form. However, if further experiments can address the questions shown below, this manuscript may be potentially acceptable.

Answer 2-1: We thank you for the thoughtful and detailed evaluation of our manuscript. Your recognition of the importance of our research is highly appreciated. We also appreciate your constructive feedback and have considered your comments when we revised the manuscript.

Comment 2-2. The major problem is that no experimental evidence confirms the formation of disulphide bond, which instead is suggested by DFT calculations. However, the formation of disulfide bond is the key point of this work, so this question must be addressed.

Answer 2-2: We agree with the reviewer's observation that the Au-cysteine aggregate is highly complex. To gain more insights into the structures, we have conducted extra experiments. Unfortunately, due to the limitations of Raman spectroscopy and the tendency of the coordination polymer to easily burn, we were unable to obtain a direct visualization of the disulfide bonds (for more details, please refer to **Answer 1-7**). As for the XPS (Figure S20 in the revised manuscript), the reported -SH and disulfide bonds in literature have exhibited a wide range of binding energies (162eV~165.1eV) and can sometimes overlap. The peak centers of 2p_{1/2} and 2p_{3/2} are closely spaced, and the binding energies are sensitive to protonation, further complicating the differentiation of disulfide bonds. Techniques such as FT-IR and solid-state NMR have also failed to provide a definitive answer regarding the disulfide bonds.

As an alternative approach, we have discovered a method to directly investigate the reaction between Au(I) and cysteine or oxidized cysteine (refer to the details in **Answer 1-2**). The results suggest that: (1) multiple coordination processes may occur simultaneously, resulting in the formation of different coordination polymers. As a result, the final products are a mixture of different structures. (2) The Au(I)-oxidized cysteine species could play a significant role in the formation of coordination polymers that exhibit good chirality inversion behavior. (3) Additionally, the stereochemistry of Au(I)-oxidized cysteine and cysteine could also influence the process.

Altogether, we renounced the claim of the disulfide bond mediated chirality inversion mechanism.

Comment 2-3. Since only 2 flakes can be found among 60 samples, the DFT calculations based on the two flakes (HRTEM) may not reveal the real structure of bulky sample. On the other hand, the distance of aurophilic interactions is around 0.281 nm assigned from the TEM analysis, so does the optimized structures confirm this value?

Answer 2-3: Thank you for bringing up the potential issue of purity. The final products were indeed a mixture of different structures. This can be seen according to the TEM results as most of the flakes are amorphous, and also multiple coordination processes would occur simultaneously during the CP material synthesis, which cannot be avoided according to our controlled experiments on the stoichiometry. The discussion has been modified in the corresponding parts.

We have also re-calculated the DFT simulations. The optimized structures have confirmed the presence of 0.281 nm aurophilic interactions in the Au-cys structure. “However, when the structure was then subjected to 6.125% Au → Ag substitution, strikingly, no convergence was achieved during relaxation of the 3D-periodic lattice models. We attribute this to the changes in metal-S and metal-N distances, which increase from 2.27 Å and 2.16 Å for Au-S and Au-N to 2.35 Å and 2.20 Å for Ag-S and Ag-N, respectively.” Thus, we removed the previous discussions on the AuAg_{0.1}-cys structures. The Au-cys simulation results are maintained since we have obtained the lattice fringes of such CP materials for the first time, and the simulation results might be helpful for others in the future.

Comment 2-4. To track structural changes and confirm the optimized structures, UV-Vis and CD spectra of Au-S bonds and aurophilic interactions are not enough, the bonds such as Au-N, Au-O are also important. Because HAuCl₄ is used, the amino group of cysteine may be protonated and thus does not coordinate to Au ion.

Answer 2-4: The reviewer is correct.

The pH of the growth system is important to the formation of coordination polymers. For cysteine, the pK_a(-COOH) is 1.96, pK_a(-NH₃⁺) is 8.18, pK_a(-SH) is 10.29. At pH of 1, the -NH₃⁺ and -COOH would all be protonated, and the obtained products showed distinct optical features. At pH of 3, the -NH₃⁺ would still be protonated, but coordination polymers with chirality inversion behavior could be obtained. Higher pH is not detrimental to the chirality inversion behavior, but mainly influences the shape of g-factor plots.

Figure R9. The responsive behavior of L0.06 CP materials prepared at different pHs.

Comment 2-5. I notice that the authors use HAuCl_4 to synthesize samples, which indicates that redox reaction may take place during synthetic process, so the sample may be a mixture. Is it possible the cysteine -SH group be oxidized to C=S or -S-S- bonds?

Answer 2-5: Thanks for your comments. The reaction between HAuCl_4 and cysteine has been previously carefully explored. Their results suggested -S-S- disulfide bonds would be formed from the cysteine -SH group. The -S-S- disulfide bonds could be further oxidized to $-\text{SO}_3\text{H}$ with Au(0) nanoparticle catalysts in alkali conditions.

Figure R10 (a) Reaction mechanism between HAuCl_4 and cysteine; (b) ESI-MS spectra of the products obtained after the reaction. Left: negative ion mode; Right: positive ion mode. (Reproduced with permission from *Dalton Transactions* 2014, **43**, 3911-3921, Copyright 2014 Royal Society of Chemistry)

Comment 2-6. The optimized structure indicates that the valence of Au is +1. Does the EXAFS spectra exclude the Au^{3+} ion?

Answer 2-6: Thank you for the kind comments. The EXAFS spectra of L0 and L0.06 showed different features with the AuCl_3 . The XPS results further exclude the Au^{3+} ions.

Figure R11 (a) Enlarged area of the pre-edge of metallic Au, L0, L0.06 and AuCl_3 ; (b) XPS Au 4f high resolution scan of L0 and L0.06. The binding energies of Au ($4f_{7/2}$ L0: 84.35eV; L0.06: 84.05eV) are higher than those of the typical metallic Au (83.95 eV, *Surf. Interface Anal.* 1998, 26, 642 (1998)) and lower than the HAuCl_4 (85.70eV, *Science*, 2020, 368, 642-648), and close to those reported Au(Ag)-cysteine coordination polymers.

Comment 2-7. In the synthesis section, the AgCl precipitate was not removed, which apparently influenced the results obtained from the energy dispersive X-ray spectra.

Answer 2-7: Since we used CTAB-decanol surfactant systems, the Ag⁺ is more possible to form AgBr₂⁻ complex. Nonetheless, the AgBr₂⁻ would further form complex with cysteine, since the cumulative formation constant for AgBr₂⁻ are 2.1*10⁷, while the k_{sp} of Ag₂S is 6.3*10⁻⁵⁰ (formation constant of Au-cys is not found in the Lange's Handbook of Chemistry).

The composition of the AuAg_{0.06}-Lcys (L0.06) was further researched by XPS (Au : Ag=1:0.045) and ICP-OES (Au: Ag=1: 0.056), which are similar to the results obtained from EDX results (Au: Ag=1:0.085)

Reviewer #3 (Remarks to the Author):

Comment 3-1: Incorporation of foreign metals in nanomaterials enhances several of their properties such as catalytic activity, optical and chiroptical properties; hence, it is of high importance for tuning their properties and broadening the scope of applications. In most of the cases, enhancement in specific properties was observed upon alloying due to the synergistic effect. In particular, silver-doping was used as a strategy to enhance the optical properties of helical gold-cysteine supramolecular assemblies. [see DOI : 10.1039/c9cp00829b] This seminal work published in PCCP, presenting exhaustive characterizations (while I must confess here that high-resolution transmission electron microscopy has allowed the authors to gain insights into the lattice for the first time, partially guiding the use of DFT calculations to support suggested crystal structure models at full atomic detail) and also top-level DFT and TD-DFT calculations to figure out the key ingredients in the enhancement effects, is cited in the present manuscript but never in the manuscript it is mentioned that silver-doping was used before as a strategy to enhance the optical (both nonlinear and chiroptical) properties. Thus the novelty is significantly reduced since PCCP was already dealing with helical silver-doped gold-cysteine supramolecular assemblies. Concerning the reversible chirality inversion of an AuAg_x-cysteine (AuAg_x-cys) coordination polymer by simple pH changes, I must confess that it is a quite appealing strategy.

Answer 3-1: Thank you for your valuable input. Based on your constructive comments, we have made significant efforts to refine and strengthen the discussions and evidence including integrating more characterization and controlled experiments to support our argument. Your feedback, along with that of other reviewers, has greatly improved our manuscript.

We would like to acknowledge the significant contributions of Rodolphe Antoine, as mentioned in PCCP paper of Ref. 20 (now ref. 9 in the revised manuscript), who has extensively studied the **non-linear optical properties of Au-cys and Ag-doped Au-cys coordination polymers**. Their work involved careful optical measurements, such as UV-Vis spectrometry, fluorescence spectrometry, CD spectrometry, and two-photon excited fluorescence spectrometry. In addition, top-level DFT and TD-DFT calculations were employed to investigate the origin of the optical properties. The study considered different model systems for supramolecular assemblies, namely M₃-Cys₄-CH₃ with M = Au₃ and Ag-Au₂, and M₅-Cys₆-CH₃ with M = Au₅ and Ag-Au₄, to understand the role of silver atoms and the transition from smaller to larger subunits, where helical structures are expected to form. This work, along with other publications by Rodolphe Antoine (Ref. 20, *Prog. Nat. Sci. Mater.* 2016, 26, 455-460, now ref. 8 in the revised manuscript), provides comprehensive references for understanding the optical properties of Au(Ag)-cys materials.

In our manuscript, our focus is primarily on the pH-induced **chiral responsive behavior** of the AuAg_{0.06}-cys (L0.06) coordination polymers. We investigated the chirality inversion behavior, methods to enhance the reversion efficiency, the pH switching point, the possible structures of L0.06, the reaction kinetics and stoichiometry, etc. While

CD spectrometry is a key instrument for studying chirality inversions, our focus was not on the origin of the CD peaks, but rather on the signs of the peaks and attempting to identify the peaks with the possible structures (Au-S bonds and Au-Au(Ag) interactions) based on published results. Therefore, the novelty of the work published in PCCP does not affect the novelty of our study. It is true that the PCCP paper has already proposed the existence of helical silver-doped gold-cysteine supramolecular assemblies and their optical properties. However, the chirality inversion behavior of Au-cys, Ag-cys, and AuAgx-cys coordination polymers, which we have explored in our study, has not been previously investigated.

Comment 3-2: But once again, I am concerned by the novelty of this strategy, since reversible chiral transformation in supramolecular helices was already reported in DOI:10.1038/s41598-018-29152-9 where acidic environments shift the helicity to left-handedness while the alkaline conditions reversed the helical structures to righthandedness. Also Chirality reversal, enhancement and transfer by pH-adjusted surfactant assembly was reported in DOI: 10.1039/D0CC07008D

Answer 3-2: We agree with the reviewer that chirality inversions have been previously observed in other studies. The paper (Sci. Rep. 2018, 8, 11220, 10.1038/s41598-018-29152-9) reports the synthesis of naphthalenediimide appended L-glutamate (NDI-L-Glu) that self-assembles into chiral supramolecular structures. The structure is sustained by π - π interactions and hydrogen bond networks. The optical activities of such materials could be symmetrically reversed by pH in the range of -5 mdeg to +5 mdeg. The paper (Chem. Commun., 2020, 56, 15345—15348, 10.1039/D0CC07008D) reports the pH-regulated TD-AlaA selfassembly process and found that its optical activity can be reversed. The reversion of the optical activity marked by CD spectra was not symmetric and only the signs of a part can be reversed. Overall, inverting the chirality has been observed by modulating the supramolecular assembling helical superstructures in some cases, triggered by light, temperature, pH, etc. The structures are typically sustained by π - π interactions and/or hydrogen bond networks. In our case, the structures are mainly sustained by the -Au-S- coordination chains and the Au-Au/Ag aurophilic interactions, and the CD peaks could be reversed in the range of -1100 mdeg to +600 mdeg, and the g-factors could be reversed in the range of \sim -0.04 to \sim +0.03, which are quite impressive as well. We have included such discussion in the introduction part.

“...On a larger scale, the advancement in nanoscience recently has kindled new fires in chirality^{4,16}. Chirality inversion has been observed by modulating the supramolecular assembly of helical superstructures in certain cases¹⁷⁻²⁴, triggered by factors such as light, temperature, pH, coordination reactions, etc. The building blocks generally involve a combination of chiral units and chromic molecules and the assembling structures are typically sustained by π - π interactions, coordination, electrostatic forces, and/or hydrogen bond networks^{22,25}. Stimuli on the responsive functional groups within the building blocks could induce the tuning of the supramolecular assembling structures, resulting in the chirality responsive behavior...”

Comment 3-3: Although the silver doping strategy and the reversible chiral transformation in supramolecular helices are certainly the most remarkable results of this work, given the pioneering work cited above, I am sceptical about the novelty criterion that justifies publication in Nature Communications.

Answer 3-3: Thanks for bringing the papers above to our specific attention. We have included and added more discussions of those papers in the text. Considering the focus of those papers and ours, we believe the novelty of our results will not be influenced, and our results could rich the concept of chirality inversion systems. We have rewritten the introduction part accordingly.

Comment 3-4: Again screening several different surfactants in the synthesis, which typically could influence the polymerization process and therefore affecting the chiral properties is not new. Indeed, surfactant-induced chirality

transfer, amplification and inversion was already reported in cucurbit[8]uril–viologen host–guest supramolecular system (DOI: 10.1039/D1TC03975J).

Answer 3-4: The reviewer is correct. Screening the surfactants is very common, so we didn't pay much attention to the methods themselves, but to the materials we have obtained. We specifically focused on the properties of AuAg_{0.06}-L-cys (L0.06) prepared in the CTAB-decanol surfactant systems.

Comment 3-5: That said, I encourage the authors to better evaluate the novelty of their work in Introduction based on the reported seminal work on silver doping strategy and reversible chiral transformation in supramolecular helices.

Answer 3-5: This point is well accepted. We have modified the introduction part and included more discussion on the reported chirality inversion supramolecular systems, the unique features of Au-cys coordination polymers, as well as the doping strategy. We started with a general discussion on the chirality-endowed uniqueness of Au-cysteine (Au-cys) coordination polymers (CPs), which lead to that responsive chiral behavior might be more interesting. Then we offered the background of the state-of-art of responsive chiral systems. By considering the structures of Au-cys CPs and the exposure of function groups, it might be promising to explore the responsive behavior of Au-cys CPs. At the end of this part, our new findings were revealed.

Comment 3-6: For these reasons, I request that this article be transferred to Communications Chemistry or Scientific reports.

Answer 3-6: We appreciate your constructive feedback and have considered your comments when we revised the manuscript. We hope the revised manuscript could meet your expectations.

Other comments:

Comment 3-7: Ref. 18 (DOI: 10.1039/C3SC22215B) does not present any chiroptical measurements neither d= silver-doping results, Thus the sentence “The peaks in the CD spectrum of L0.06 from 300nm to 450nm, which arise from the Au-Au/Ag AuroIs[18], showed a symmetric inversion, while the CD spectrum from 210nm to 300nm[18]” should be better argued. The strong CD band from 300nm to 450nm may likely found its origin in a geometrical structure, which couples together the S-Au bond localized LMCT transitions in a chiral way. (see DOI: 10.1016/j.pnsc.2016.08.008).

Answer 3-7: Thank you for pointing this out. We have modified the corresponding references. Ref. 18 (*Chem. Sci.* 2013, 4, 1852-1857, now ref. 7 in the revised manuscript) is more about the absorption peaks. Ref 19 (*Prog. Nat. Sci. Mater.* 2016, 26, 455-460, DOI: 10.1016/j.pnsc.2016.08.008, now ref. 8 in the revised manuscript) and Ref 16 (*Science*, 2020, 368, 642-648, now ref. 5 in the revised manuscript) have treated the CD spectrum.

Comment 3-8: Fig S5 should report data on pristine coordination polymers (L0 I guess?). Thus why does the caption mention L0.2? In the same vein, does surfactant was added to L0 solutions, results for alkaline conditions?

Answer 3-8: Fig. S5 shows the responsive behavior of Ag-doped Au-cys coordination polymer in acidic conditions. We have now changed it into the g-factor plots and UV-Vis spectra of the L0.06 prepared in CTAB-decanol surfactant systems. The results suggested that the pristine L0.06 will not change under acidic condition.

Figure S5 CD and UV-Vis spectra of L0.06(prepared in CTAB-dec surfactant system) in HCl solution

Our doping was directly achieved via adding HAuCl_4 together with AgNO_3 at the beginning of the reaction. The number in the abbreviations indicated the Ag/Au value. For example, L0 means no AgNO_3 was introduced in the synthesis, L0.06 means a ratio of HAuCl_4 : $\text{AgNO}_3=1:0.06$ was introduced in the synthesis. All the CP materials could be synthesized with or without surfactants. Since the CTAB-decanol solution offer a good condition to prepare CP materials with good chirality inversion behavior, from the second paragraph of the **Results** part, all the syntheses were conducted in the CTAB-decanol solution, no matter L0 or L0.06. The method is close to that reported in *Science*, 2020, 368, 642-648, but different from that in the *Phys. Chem. Chem. Phys.* 2019, **21**, 12091-12099 where a two-step method is used. For L0, no AgNO_3 solutions was added in the synthesis, while CTAB-decanol surfactant system was still used. The responsive behavior of L0 CP material is shown in Figures S13.

Figure S13 Responsive behavior of L0 in different pHs. “cy” in the legends indicates cycle, and “neu” denotes neutralization. The neutralization was checked by pH paper. No obvious inversion was found.

Comment 3-9: The authors should better discuss the similarity/difference observed with spectra reported in Fig. S7 between pristine and silver doped supramolecular assemblies with the CD and UV/vis spectra reported on the seminal paper DOI : 10.1039/c9cp00829b (to better understand the shift observed as a function of doping level in spectra).

Answer 3-9: Thanks for the kind suggestion. The Ag doping could effectively modulate the UV-Vis absorptions and

the CD spectrum. The PCCP paper nicely explains the effect of Ag doping on the two-photon absorptions (Figure R11a). The *Prog. Nat. Sci. Mater.* 2016, **26**, 455-460 paper suggests that the strong CD band from 300nm to 450nm could arise from the S-Au bond localized LMCT transitions coupling in a chiral way via aurophilic interactions (Figure R11b). The *Science*, 2020, 368, 642-648 paper shows that the CD spectrum is highly sensitive to the twisting morphologies of the CP materials (Figure R12c)

Figure R12 (a) one-photon absorption (OPA) and two-photon absorption (TPA) of Au₄-Ag-Cys₆-CH₃ (left) and Au₅-Cys₆-CH₃ clusters. It can be seen that replacing one Au by Ag will not significantly change the OPA, but influence the TPA a lot. Reproduced with permission from *Phys. Chem. Chem. Phys.* 2019, 21, 12091-12099, Copyright 2019, Royal Society of Chemistry. (b) Calculated ECD spectrum (sticks: rotatory strengths versus wavelength, lines: Lorentzian lineshape, 10 nm half-widths) of the lowest-energy Au-S-Au-S-Au conformer (see below) indicating the CD activity of the metal-sulfur bond (red and blue lines and sticks indicate the spectra calculated for structures with D and L-cysteine, respectively). The analysis of the nature of the two lowest-energy excited states (marked with circle and square, D-cysteine) is given below as occupied and virtual natural transition orbitals. Reproduced with permission from *Prog. Nat. Sci. Mater.* 2016, 26, 455-460, Copyright 2016 Chinese Material Research Society. Published by Elsevier B. V. (c) Calculated CD spectra for the Au-L-Cys nanoplatforms with different twisting conformations. Reproduced with permission from *Science*, 2020, 368, 642-648, Copyright 2020, The American Association for the advancement of Science.

Since the optical properties rely heavily on the structures, and Ag-doping could effectively tune the structures and morphologies of the Au-cys CP materials, it might not be rigorous to directly compare our results with the reported data. Thus, only a descriptive discussion is provided in the revised manuscript. It is also worth noting that the Ag-doped Au-cys CP materials obtained in different papers showed different optical properties with different peak centers, peak shapes, peak intensity (Figure R13), suggesting that the properties are highly sensitive to the synthesis conditions or product structures.

Figure S12 UV-Vis spectra and g-factors of $\text{AuAg}_x\text{-cys}$ coordination polymers prepared in CTAB-decanol solutions with different amounts of Ag doping. The numbers in the figure legends indicate the molar ratio of Ag to Au.

According to (a), the Au0Is of D0 displayed a broad peak from ~ 300 to ~ 400 nm. The main peak resides at around 355nm, while a shoulder peak resides at around 320nm. When the amount of Ag increased, the main peak center would gradually red-shift to 382nm for D1, while the shoulder peak becomes less obvious. Such peak is an indicative of the aurophilic interactions.

According to (b), the optical activities changed more significantly with the increase amount of Ag. For $\text{AuAg}_x\text{-D-cys}$ materials, the signs of g-factors are negative from ~ 230 nm to ~ 350 nm, and show a positive peak at from ~ 350 nm to ~ 400 nm. The positive peak would gradually blue-shift from 388 nm to 360 nm with the increase of Ag doping, and the intensity would first increase and then decrease. Such peak might originate from a geometrical structure, which couples together the S-Au bond localized ligand-metal charge transfer transitions in a chiral way via aurophilic interactions. The shape of the negative part showed an intensity increase as the Ag doping increases.

Figure R13 (a, b) The optical properties of the $\alpha\text{-Au-Cys}$, $\beta\text{-Au-Cys}$, and Ag-doped Au-cys materials prepared in *Phys. Chem. Chem. Phys.* 2019, 21, 12091-12099; Reproduced with permission from *Phys. Chem. Chem. Phys.* 2019, 21, 12091-12099, Copyright 2019, Royal Society of Chemistry. (c) The optical properties of the Au/Ag-D(L)-Cys prepared in *Science*, 2020, 368, 642-648. Reproduced with permission from *Science*, 2020, 368, 642-648, Copyright 2020, The American Association for the advancement of Science.

(d) The optical properties of our L0 and L0.06 CP materials.

Comment 3-10: Of note, these results indicate that the structures of the obtained AuAg_x-cys coordination polymers are Au-S covalent chains intertwined by intra- or inter-chain linear Au-Au/Ag AuroI_s to form amorphous nanosheets, and the thickness of the sheets was 1.2nm. This thickness is similar to the thickness reported by Attila Bóta and coworkers (<http://dx.doi.org/10.1016/j.colsurfa.2015.01.048>) on supramolecular chemistry of gold and L-cysteine. Looking at STEM images of the L0.06, I am thus a bit confused, these images look more similar to “beta-sheet” than “alpha-helix” motifs. Also how realistic are the structural motifs found by DFT calculations as compared to experimental data. Why not trying beta-sheet motifs for calculations ?

Answer 3-10: Thanks for the insightful comments. The STEM images (Figure 3a-d, S16) can only show 2D projections of the nanomorphology and do not provide atomic-resolution. Therefore, deducing the atomic structures is challenging. The thickness can be roughly measured based on the standing sheets (Figure 3b), which is close to 1.2 nm. This value is similar to the thickness of the so-called β-Au-cys structure reported by Attila Bóta (Ref. 17, *Colloids Surf. A Physicochem. Eng. Asp.* 2015, 470, 8-14, now ref. 6 in the revised manuscript). They proposed Au-cys coordination polymers with lamellar structures. However, in our case, the entire sheet is amorphous and appears as intertwined wires (Figure 3c, d), which are not as uniform as the proposed structures in Attila Bóta's paper. Therefore, we are unsure whether our structures resemble the α-Au-Cys or β-Au-Cys structures proposed in Attila Bóta's paper. Since we obtained lattice fringes from 2 flakes of the D0.06 samples, we created a series of Au-cys unit cell candidates that would reproduce the observed distances and cell vectors. After eliminating structure models of unrealistic atomic overlap, the proposed structure was refined from DFT calculations (see Methods). The resulting Au-Cys unit cell as illustrated in **Figure S22** is similar to the reported lamellar crystal structure (both Attila Bóta's paper and N A Kotov's paper).

Comment 3-11: How do these calculated structures compare with those reported in DOI: 10.1039/c9cp00829b (which nicely explain silver doping effect in optical properties of gold-cysteine supramolecular assemblies).

Answer 3-11: In Ref 20 (now ref. 9 in the revised manuscript), two model systems for the supramolecular assemblies have been considered, namely M₃-Cys₄-CH₃ with M = Au₃ and Ag-Au₂ and M₅-Cys₆-CH₃ with M = Au₅ and Ag-Au₄ (Figure R12a). The calculations nicely explained the silver doping effect in the one-photon absorption and two-photon absorption. The models are based on oligomers and aimed at understanding the origins of the optical properties. In our cases, the calculation is based on crystal structure with periodic lattice. The model is more like those shown in *Science*, 2020, 368, 642-648 (now ref. 5 in the revised manuscript). We optimized the Au-cys crystal structure according to the lattice fringes we have obtained. Then the structure was then subjected to 6.125% Au → Ag substitution, strikingly, no convergence was achieved during relaxation of the 3D-periodic lattice models. We attribute this to the changes in metal-S and metal-N distances, which increase from 2.27 Å and 2.16 Å for Au-S and Au-N to 2.35 Å and 2.20 Å for Ag-S and Ag-N, respectively. Since our DFT calculations are bound to 3D-periodic unit cell models, the structural insight from such calculations can only be qualitative.

Comment 3-12: More comments on the relative ratio of alpha vs beta motifs (that would lead to completely different optical properties see DOI: 10.1039/c9cp00829b) should be added.

Answer 3-12: We agree with the reviewer that obtaining more results on the possible ratio of different structures inside the AuAg_x-cys coordination polymers (L0.06 for example) would greatly contribute to understanding the materials. However, we are uncertain whether there are clear α-Au-Cys and β-Au-Cys motifs present in our AuAg_x-cys materials as the Ag doping could significantly tune the structures including the nanomorphologies and the

conformations, and we do not have a method to quantify their presence. As mentioned in Ref. 20 (now ref. 9 in the revised manuscript), the incorporation of Ag into the structures can alter their composition, as demonstrated by charge-detection mass spectrometry. Additionally, our lattice fringes suggest the existence of other potential structures. According to our new experiments on the stoichiometry, **at least 3 different kinds of L0.06 CP materials with different optical properties and responsive properties can be obtained:** (1) the one with chirality inversion behavior (CP1); (2) the one, which only the CD peak at $\sim 370\text{nm}$ can be inverted; (3) the one with no intense absorption and CD peak at around 370nm . Therefore, we lack confidence in analyzing the ratio between different potential structure motifs.

Comment 3-13: I agree that according to my knowledge, this is the first time that obtaining the fringes of Au(Ag)-cys coordination polymers is reported. However, the purity is not really controllable and satisfactory (only 2 flakes of the D0.06 sample showed crystalline fringes over more than 60 different coordination polymers).

Answer 3-13: We appreciate your recognition of our results. It is true that the obtained coordination polymer materials consist of a mixture of different structures. The lattice fringes might also be helpful for other studies, thus although we didn't get the DFT simulation optimized structure of AuAg_x-cys CP, we kept the structure of Au-cys CP in the revised manuscript. We have made modifications to reflect this in the relevant sections.

Comment 3-14: Doping the coordination polymers with a proper amount of Cu cations (and other metal cations) to tune chirality behavior and possibly chirality switching behavior is interesting and should be better highlighted since here is the real novelty of doping strategy (even if the effect of metal doping is not at the level of silver doping). For instance EDX and SAED mapping and/or XPS might permit to better evaluating the level of doping using metal ions other than silver.

Answer 3-14: Thank you for your kind comment. Cu-cys oligomers or coordination polymers have been reported previously (*Phys. Chem. Chem. Phys.*, 2023, 25, 6025-6031; *Inorg. Chim. Acta*, 2009, 362, 395-401). And the Cu doping was also tested in *Science*, 2020, 368, 642-648 (now ref. 5 in the revised manuscript) to tune the structures. Here we have added more characterizations and discussions on the Cu-doped materials (Figure S27).

Figure S28. Further structural characterization of the Cu-L0.02

(a, b) HRTEM image and the corresponding SAED patterns. The patterns suggested the amorphous nature of the product.

(c, d) STEM images show the layered structures and the small dots around the layers.

(e - h) EDX mapping of the Cu-L0.02 particles.

Reviewer #4 (Remarks to the Author):

Comment 4-1: As a computational chemist, I will focus on the theoretical aspects of the manuscript, relying on other Reviewers to assess the experimental part. In this manuscript, the authors report a pH-controlled chirality inversion of an Au-cysteine coordination polymer upon doping with Ag. The work is thorough and includes a screening of different ligands and cations, which highlights the unique features of Ag and cysteine. Although the operating mechanism is not very clear, the phenomenon per se is quite interesting and could trigger new concepts and ideas in the field of inorganic chirality. After addressing my comments below, I would recommend publication in Nat. Commun.

Answer 4-1: We thank the reviewer for this positive assessment. The requested improvements are listed below.

Comment 4-2: -The computational details are incomplete; the authors must indicate the density functional, the dispersion correction, and the K-points. Please also indicate that cell relaxation was performed (Figure S19d,e).

Answer 4-2: We thank the reviewer for pointing the missing details out. We have added the missing information in the methods section.

“All calculations and relaxations were performed using the PWscf code of the Quantum Espresso software package^{52,53}. Atomic cores were represented by norm-conserving Vanderbilt pseudopotentials with a plane wave basis cutoff energy of 50 Ry using the PBE functional with Grimme D3 dispersion⁵⁴⁻⁵⁶. Gamma point calculations were used to obtain the structures and energies. Visualization and system setup was done using Vpster⁵⁷. Initial starting structures were estimated using the TEM data. Optimization of the Au-Cys unit cell models was performed in multiple steps. We first kept all heavy atoms fixed whilst refining only the positions of hydrogen atoms. Next the remaining atomic positions were relaxed before finally the entire structure, including the cell edges were optimized to lowest energy, see also Figure S22. The final Au-Cys cell was then explored for Au → Ag substitution. For this, randomly depicted Au atoms were replaced by Ag yielding a AuAg_{0.06125}-Cys unit cell model, which was relaxed in analogy to the Au-Cys models. Strikingly, upon such substitution no convergence was achieved during relaxation of the 3D-periodic lattice models.”

Comment 4-3: -Inputs and direct coordinates of the structures must be fully available in the SI for visualization and reproducibility.

Answer 4-3: The input files, including the direct coordinates, of all calculations whose data is used in the manuscript are now collected in the pw_inputs.zip file to be made available in the SI.

Comment 4-4: -Figure 4. In (a), all cysteines have NH- and COOH groups. In (b), some cysteines present NH₂ and COO- groups. Could the authors comment on these structural changes as well? Was a NH₂/COO- configuration ever evaluated for (a) (i.e., without Ag)?

Answer 4-4: The protonation status, NH₂ and COOH, were the same for all calculations presented in the manuscript. We have provided additional images from different angles to improve clarity. We also conducted further relaxations based on different protonation status. These did either not converge, lead to structures incompatible with the HRTEM cell vectors, or were energetically disfavored.

Comment 4-5: -How many Au atoms are in the DFT-optimized Au-Cys unit cell? When one Au is substituted by Ag, how much Ag% is present in the unit cell? By looking at Figure 4b, it seems more than the experimental 10%. Have the authors considered using a larger Au-Cys unit cell (i.e., duplicate one direction) to dilute the Ag atom?

Answer 4-5: In the initial manuscript the Au-Cys cell included 10 Au atoms, and the Au-Ag-cys cell substituted 1 Au atom with 1 Ag. However the correct amount of Ag would be 6%. We have thus taken the time to replicate the calculations based on a larger cell with 32 Au atoms in the Au-Cys cell. In the Au-Ag-Cys cell 2 Au atoms have been substituted with Ag yielding an Ag% of 6.125.

Comment 4-6: -In Figure S20d,e, the authors compare experimental vs. simulated EXAFS of L0.06. For completeness, I would recommend performing the fitting of L0 as well.

Answer 4-6: Thanks for the suggestion. EXAFS is a short-range order analysis technique up to a very few coordination spheres. The fitting of the EXAFS data could in theory give the coordination numbers and bond length. However, if the material is amorphous and there exist multiple potential coordination structures in the spheres, the fitting would be quite difficult and sometimes cannot give reasonable results. As we recalculated the optimized structure and failed to obtain the convergent AuAgx-cys structure, although the L0 and L0.06 spectra can be fitted as well, we feel it might be better to remove the EXAFS results from the current manuscript since our products are mainly amorphous.

Comment 4-7: -Suggested literature on DFT of monometallic Au/Ag with cysteine, DOI: 10.1039/D0RA06486F

Answer 4-7: Thank you for bringing this to our attention. We have added this reference as ref. 44 in the revised manuscript.

Reviewer #5 (Remarks to the Author):

Comment 5-1: In this manuscript, the authors report the dynamic switching of CD signals of AuAgx-cysteine assemblies by tuning pH values. The reversible switching of CD signals of supramolecular assemblies is very common, however, it has not been well-demonstrated in Au(I)-thiolate coordination polymers, and unrevealing the underlying mechanism is important. After carefully reading it, I find out there are three major drawbacks of this work:

Answer 5-1: Thank you very much for your thorough review of our paper. We greatly appreciate your positive feedback and the value you find in our work. We are grateful for your comments on the structures and mechanisms discussed. Furthermore, we have carefully considered your suggestions and made the necessary revisions to the manuscript in order to address any concerns you have raised.

Comment 5-2: the first is that the authors did not provide sufficient experimental data to test the proposed structural model of AuAgx-cysteine assemblies obtained by DFT simulation, and it is unclear the overall packing structure of the assemblies from the given data;

Answer 5-2: Thanks for the insightful comment. In our revision, more characterizations were conducted to shed light on the structures of AuAgx-cysteine CP materials, such as Zeta-potential titrations, Raman spectrum, XPS, ICP-OES, solid-state NMR, vibrational CD spectrum, Raman optical activity, etc. However, due to the complexity of the AuAgx-cysteine CP materials and the instability under certain conditions (such as laser), an unambiguous structure cannot be clearly obtained. Thus, we re-optimized the previous DFT simulations. The updated simulation results suggested that the undoped Au-cysteine CP materials have a lamellar structure, which is similar to the reported results.

However, when substituting part of the Au atoms with Ag atoms, no convergence was achieved during relaxation of the 3D-periodic lattice models. This phenomenon in another way suggests that doping Ag to the Au-cys CPs can drastically tune the structures. Although the failure to obtain a clear structure hinders the understanding of the chirality inversion mechanism, we successfully obtained more essential information on the chirality inversion behavior through the **reaction kinetics** and the extensive **stoichiometry study**. The kinetics suggested that the covalent bond formation was rapid and then the conformation of the CP materials would continuously evolve. The reaction stoichiometry investigation showed that the formation of CP materials with chirality inversion behavior requires the balancing between different coordination and polymerization processes. **Based on these results, we renounce the claim that the disulfide bond holds the key to the chirality inversion mechanism.** We must admit that although the oxidized cysteine could play a significant role in the CP material formations, the results could not support the disulfide bond mediated inversion mechanism. As a result of this revision, the manuscript becomes more structured and credible in its argumentation.

Comment 5-3: the second is that the authors did not provide reasonable explanations on how the dynamic stereochemistry of disulfide bridge can change with pH values to result in the spectral change, actually, they even did not provide any experimental evidence that disulfide truly exist in their assemblies;

Answer 5-3: We have included additional characterizations, such as Raman spectra, XPS spectra (Figure S20), and solid-state NMR (Figure S21), in an attempt to investigate the presence of disulfide bonds. However, due to limitations in the materials and techniques, we were unable to directly observe the disulfide bonds. Nevertheless, we conducted a series of controlled experiments on stoichiometry, which provided evidence supporting the incorporation of oxidized cysteine moieties into the structures.

Comment 5-4: the third is that the authors suppose the inversion of CD signals is related to the change of stereochemistry of the disulfide bridge, but they did not explain why Au-cysteine assemblies without dithiol bridges have very similar UV-vis and CD spectra to those of AuAgx-cysteine assemblies with dithiol bridges.

Answer 5-4: The UV-Vis and CD spectra of Au-cysteine (L0) and AuAg0.06-cysteine (L0.06) are presented in Figure R13. It is evident that their UV-Vis spectra exhibit some similarities, but their CD spectra are distinctly different. The absorptions below 250nm and around 270nm in the UV-Vis spectra are attributed to the Au-S bonds, while the absorptions at approximately 350nm and 378nm are influenced by two-photon absorptions and arise from aurophilic interactions.

Regarding the CD spectra, the prominent CD band spanning from 300nm to 450nm likely originates from a geometric structure that incorporates chiral ligand-metal charge transfer transitions through aurophilic interactions facilitated by the S-Au bonds. The stereochemistry of the system plays a crucial role in determining these features. As a result, the CD spectra of L0 and L0.06 exhibit markedly different intensity characteristics, and the signs of g-factors in the 400nm to 550nm range are opposite, with L0 showing negative signs and L0.06 displaying positive signs.

Figure R13. The optical properties of L0 and L0.06.

Comment 5-5: Because of the insufficient characterizations of the assembled structure and the lack of understanding of the relationship between the assembled structure and CD signals in this work, I do not think the work can make a significant contribution in the field of chirality, so I cannot recommend it for publication in current stage.

Answer 5-5: We appreciate your valuable comments and constructive criticism, and we have tried our best to make necessary revisions to the manuscript to improve its quality.

Comment 5-6: The authors states that “inverting the chirality of a material is of great interest, as it could provide a novel way to utilize chirality in material science, however, it is also extremely challenging.” Actually, there are many reports on dynamic inversion of CD signals in supramolecular assemblies (several examples: *Angew. Chem. Int. Ed.* 2016, 55, 2994-3010; *Angew. Chem. Int. Ed.*, 2019, 58, 785 -790; *Chem. Sci.*, 2022, 13, 13623-13630), and these related work should be included in the introduction.

Answer 5-6: Thanks for bringing these nice papers to our attention. We have included these as Ref. 17, 18, 21 in the introductions.

Comment 5-7: The authors claim their assemblies are amorphous, then why they can show lattice fringes. Are the assemblies pure or composed of different types of structures?

Answer 5-7: Thanks for pointing this out. The obtained coordination polymer material is a mixture of different structures. While most of the flakes are amorphous, 2 of the screened 60 D0.06 flakes showed lattice fringes. Then we tried to dig some info from the fringes. However, since we failed to obtain a clear AuAgx-cys CP structure, we have removed the discussions on the crystal structures as well as the related mechanism interpretation.

Comment 5-8: The illustration of the packing structure should be clearer. It’s very difficult for readers to understand the structure of the assemblies in Fig. 4. And the authors are suggest to provide experimental data to support such a structural model, such as the existence of disulfide and Au-N bonds in AuAgx-cysteine assemblies.

Answer 5-8: Thanks for your suggestion. Even though we conducted much more experiments and tried a lot to explore the potential structures, we failed to obtain direct and solid evidence on the AuAgx-cys structure. Thus, the corresponding discussion has been removed.

Comment 5-9: The authors are suggested to use more characterization techniques (such as IR, XPS, solid state NMR, et al.) to elucidate how the assembled structure changes before and after CD signals are inverted.

Answer 5-9: Thanks for the helpful suggestion. We attempted two methods to isolate the materials after the CD signal reversion. The first approach involved centrifugation after the alkalization process. However, upon redispersion and immediate measurement of the CD spectrum, we observed that the g-factor plot reverted back to its original state, albeit with a slight change in peak shape (Figure R14). Consequently, we lack confidence in this method for obtaining the isolated products suitable for further characterizations.

Figure R14 Comparison between the original L0.06 optical properties and the redispersed alkalized L0.06

The second method involved directly drying the product in KOH solutions. However, the increasing concentration of KOH during drying posed challenges. Nevertheless, we managed to obtain some products in the end and proceeded with XPS measurements. Unfortunately, the analysis of the peaks proved to be quite complex, with several unusual peaks observed (Figure R15). As a result, extracting key information from such spectra posed difficulties. Additionally, we attempted solid-state NMR analysis, but obtaining enough dried product was challenging. Therefore, we only have solid-state NMR results for L0 and L0.06, which suggested similarities in hydrogen atom bonding but significant differences in the chemical environments of carbon atoms (Figure S21).

Figure R15 The XPS spectra of L0, L0.06, and the alkalinized L0.06. The C1s and Ag3d scans of the alkalinized L0.06 showed strong peaks at high binding energies, which cannot be rationally assigned.

Comment 5-10: Additional questions: How are the contents of Ag in the assemblies quantified? Why the UV spectra of AuAgx-cysteine assemblies change with pH values? Why NH₃ cannot induce the reversion of CD signals? Will OH⁻ bind with Au⁺ or Ag⁺ at the edge structure at pH value as high as 13?

Answer 5-10: The Ag content was characterized using various methods, including EDX (Table S1), XPS (Figure S20), and ICP-OES. These results agreed well.

Upon increasing the pH, the UV-Vis peak corresponding to aurophilic interactions shifted to 365nm and 378nm during alkalization and acidification. This peak is sensitive to the distance between Au-Au/Ag atoms, which may undergo slight adjustments during the response processes. However, other peaks did not exhibit significant changes. It should be noted that the shift in the UV-Vis spectrum does not necessarily lead to the reversal of CD signals, as indicated by the NH₃ experiments.

The presence of NH₃ molecules in the system can influence the hydrogen bond networks and coordinate with Ag⁺ within the polymer. According to the zeta-potential titration result, the CP materials contain unsaturated coordination sites. Thus, NH₃ molecules is possible to bind with the CP materials. However, these changes did not affect the optical activities significantly. The g-factor plots showed only slight intensity changes, while the peak shapes and centers remained unchanged. However, further alkalization of the solution with KOH (to pH~13.7) resulted in the reversal of the optical activities (Figure S9). This indicates that hydrogen bond networks and metal coordination structures are not the key to the chirality inversion. Therefore, NH₃ is unable to induce chirality inversion.

OH⁻ binding to the Au(I) or Ag⁺ ions at high pH is possible, as suggested by the coordination equilibrium and the rich structures in the reported aurophilic interactions (*Chem. Soc. Rev.* 2008, 37, 1931-1951; *Chem. Soc. Rev.* 2012,

41, 370-412), such binding may serve as the trigger for chirality inversion.

Comment 5-11: There are also misspelled words, such as “suspectable” ; the figure caption for Figure 4h is missing.

Answer 5-11: We have carefully proofread the manuscript again and changed the corresponding parts.

REVIEWER COMMENTS

Reviewer #1 (Remarks to the Author):

As expressed in my first report, the phenomena observed in this work are of great general interest and publication is recommended. This opinion is shared by most of the other reviewers, but all of them have lists of point which should be considered in a revision of the manuscript. The authors have responded in great detail to these suggestions, and most of the questions have been answered. The response explains most of the reasons for certain conclusions which were in doubt or not sufficiently precise. The revised manuscript appears to be a report which is now easier to read and to appreciate and acknowledge. I must admit that I did not have the time to scrutinize all details again, but rather concentrated on those passages where I was not happy when first reading the paper.

Reviewer #2 (Remarks to the Author):

In this revised manuscript, the authors have presented more detailed additional experiments, and based on the experimental results they have reconsidered the CP structure that corresponds to chirality conversion. To my opinion, my concerns have been addressed and the results have now well been supported by experimental data, so the manuscript is now acceptable.

Reviewer #4 (Remarks to the Author):

Although the authors have addressed my comments, the paper has changed significantly. The main goals of the previous DFT calculations were to support the hypothesis of the S-S bond and evaluate the substitution of Au by Ag. However, the former hypothesis has been removed, and the substitution could not be described due to convergence issues. The relevance of the computational part has dramatically decreased and it now plays a minor role in the conclusions. Thus, I would refer to other experimental Reviewers to recommend or reject this manuscript for Nat. Commun.

Reviewer #5 (Remarks to the Author):

In this revised paper, the author made a major change in the contents of the work according to the reviewers' comments and questions. They removed some questionable conclusions on chirality inversion mechanisms and added some new experimental data and discussion on the synthesis and structural information. By doing this, the main topic of this work shifts from the chirality inversion phenomena and the mechanisms to the chirality inversion phenomena, the synthesis and structure. Actually, the removed conclusions, if they were solid, are of importance, however, the authors removed those conclusions because they realized the products were a combination of different structures and they could not give a reasonable explanation on the chirality inversion mechanism, so the importance of this work is weakened a lot after revision. In addition to the above weakness, the new added discussion and conclusions on the growth processes of the assemblies are still not supported with direct structural characterizations, but supported with some control experiments instead, which make the conclusions less convincing.

Other remaining problems in understanding the experimental results are listed as follows:

1. The switching of the alkalization state to acidification state was found around pH 9.60. The pKa(-COOH) of cysteine is 1.96, pKa(-NH₃⁺) is 8.18, pKa(-SH) is 10.29, then the author concluded the switching point seemed to be not related to the pKa of cysteine. Actually, the pKa of a molecule will change after it polymerizes, aggregates or assemblies, and sometimes the change is significant, so the

authors should not refer to the pKa of molecular cysteine directly and assume it does not change after they assemble. The authors are suggested to directly characterize the protonation and deprotonation states of the functional groups.

2. When the authors discuss the different results with NH₃ molecules and OH⁻ as base, they states that "since NH₃ molecules and OH⁻ both could change the hydrogen bond networks in the CP materials by forming N-H-X or O-H-X hydrogen bonds, but only one could induce the chirality inversion, thus, the hydrogen bond networks within the CP materials don't play an important role in chirality inversion." Such conclusion oversimplifies the effect of bases and is not supported by necessary characterizations. Since the conclusion that the switching of the alkalization state to acidification state is not related to pKa is questionable, this conclusion is also questionable.

3. The authors state "the stoichiometry between HAuCl₄ and cysteine should be at least 1:3 to form the CP if no other reducing agent is added. This stoichiometry has been generally applied in the previous synthesis of Au-CP materials." However, it is not true. The redox and coordination reactions do not go step by step, and coordination of Au(I) with thiol can already occur while Au(III) are reduced, that is, as some Au(III) are reduced to Au(I), the generated Au(I) can compete with Au(III) for thiol to generate coordination polymer. Even when there are only two equiv. of thiol, CPs can also be produced, however, the yield will be lower than that with three equiv. of thiol. Therefore, it needs 3 equiv. of thiol to turn all Au(III) to Au(I)-thiolate CPs in principle, rather than guarantee the generation of Au(I)-thiolate CPs.

Other additional problems and suggestion in writing of the paper.

1. The introduction section is loosely structured, which should focus on the theme of the work, rather than giving different types of information without strong logic.

2. The authors are suggested to read the listed references carefully and avoid paraphrasing the work inappropriately.

3. For the repeatability of the work, please list all the used chemicals in the section of Chemicals, and list all the used characterization methods in the section of Characterizations.

Reviewer #1 (Remarks to the Author):

As expressed in my first report, the phenomena observed in this work are of great general interest and publication is recommended. This opinion is shared by most of the other reviewers, but all of them have lists of point which should be considered in a revision of the manuscript.

The authors have responded in great detail to these suggestions, and most of the questions have been answered. The response explains most of the reasons for certain conclusions which were in doubt or not sufficiently precise. The revised manuscript appears to be a report which is now easier to read and to appreciate and acknowledge.

I must admit that I did not have the time to scrutinize all details again, but rather concentrated on those passages where I was not happy when first reading the paper.

Answer: We would like to express our deepest gratitude for your valuable insights and constructive feedback on our manuscript. Your recommendations, especially on the stoichiometry, have brought clarity and precision to our work, making it more comprehensive and easier to understand. Your supportive and affirmative response, shared too by other reviewers, encourages us and enhances our confidence in the value of our work. We are pleased to know that you see the general interest in our study, and this pushes us to strive for higher quality in our future research.

Reviewer #2 (Remarks to the Author):

In this revised manuscript, the authors have presented more detailed additional experiments, and based on the experimental results they have reconsidered the CP structure that corresponds to chirality conversion. To my opinion, my concerns have been addressed and the results have now well been supported by experimental data, so the manuscript is now acceptable.

Answer: We are immensely grateful for your thoughtful and meticulous review of our manuscript. Your constructive feedback prompted us to conduct further experiments in order to support our conclusions, and we thank you for guiding our efforts in the right direction. We are glad to know that your concerns have been addressed in the revised manuscript and that you find the experimental data to support our results effectively. Your positive feedback regarding the acceptability of the manuscript offers us great encouragement.

Reviewer #4 (Remarks to the Author):

Although the authors have addressed my comments, the paper has changed significantly. The main goals of the previous DFT calculations were to support the hypothesis of the S-S bond and evaluate the substitution of Au by Ag. However, the former hypothesis has been removed, and the substitution could not be described due to convergence issues. The relevance of the computational part has dramatically decreased and it now plays a minor role in the conclusions. Thus, I would refer to other experimental Reviewers to recommend or reject this manuscript for Nat. Commun.

Answer: We greatly appreciate your time and effort in reviewing our manuscript and giving us your perspective. Your comments have been instrumental in refining and sharpening the focus of our work. The major revisions that changed our paper significantly were in response to the insightful inputs from you and other reviewers. We understand your concern about the computational part's diminished relevance and its minimal role in drawing conclusions. We regret any confusion or disappointment caused by the removal of the S-S bond hypothesis. Having acknowledged your reservation about the current version, we still believe that the manuscript, backed by rigorous

experimental data, holds significance and can contribute meaningfully to the scientific community. Thank you again for aiding us in our quest to provide a valuable contribution to the field.

Reviewer #5 (Remarks to the Author):

In this revised paper, the author made a major change in the contents of the work according to the reviewers' comments and questions. They removed some questionable conclusions on chirality inversion mechanisms and added some new experimental data and discussion on the synthesis and structural information. By doing this, the main topic of this work shifts from the chirality inversion phenomena and the mechanisms to the chirality inversion phenomena, the synthesis and structure. Actually, the removed conclusions, if they were solid, are of importance, however, the authors removed those conclusions because they realized the products were a combination of different structures and they could not give a reasonable explanation on the chirality inversion mechanism, so the importance of this work is weakened a lot after revision. In addition to the above weakness, the new added discussion and conclusions on the growth processes of the assemblies are still not supported with direct structural characterizations, but supported with some control experiments instead, which make the conclusions less convincing.

Answer: We sincerely appreciate your detailed evaluation of our revised manuscript and the insightful feedback provided. We acknowledge the changes you highlighted in our paper and understand your concerns about the shift in focus and the weight of the presented work. The decision to remove the mechanism parts was not taken lightly, but based on the comprehensive reassessment of our data and constructive suggestions from reviewers. The main issue was that the **structures of AuAg_x-cys coordination polymers (CPs) are too complicated**. They are typically **amorphous** and contain multiple possible structures (as evidenced by our controlled experiments on the stoichiometry parts, and also the reported structures). Our experimentally obtained lattice fringes are the first time to have a clear image on the lattices. Unfortunately, only two sets of fringes were obtained instead of three. If three sets of fringes can be obtained, we can create an unambiguous lattice structure according to these data without simulations. Although it might be only one polymorph of all existing possible structures, a mechanism can be proposed based on an experimentally obtained structure with more confidence. However, due to the lack of structural information, we can only employ DFT simulations to shed light on the structures. The DFT simulated Au-cys coordination polymers is similar to what have been reported. However, when the structure was then subjected to 6.125% Au → Ag substitution, strikingly, no convergence was achieved during relaxation of the 3D-periodic lattice models. We have also tried to conduct other characterizations in the revision, like Raman spectrum, XPS, solid-state NMR, vibrational CD spectrum, Raman optical activity, etc. However, these results cannot give a clear chirality inversion mechanism as well.

We concur that if the disulfide bond assisted chirality inversion mechanism could be proven solid, these assertions would have been noteworthy. **Such a mechanism has been found in other systems** (*J. Am. Chem. Soc.* 2023, 145, 5545), where the chiral organic cation can be converted to its enantiomer because of an electric field-induced shift of the S–S moiety relative to its screw axis during the ferroelectric switching. It has also been recently reported that the flexibility of disulfide bond can endow Cd-cystine nanostructures with tunable twisting angles, pitch, width, thickness, and length (*Nature*, 2023, 615, 418). Actually, in their SI Fig. S22. Role of pH in self-assembly, **the handedness of the obtained bowtie structures inverted at different synthesis environment pHs**, although they didn't discuss this effect in their manuscript. We have prepared Cd-cystine bowtie structures according to their protocol and studied their responsive behavior at different pHs during our first-round revision. However, the Cd-cystine would decompose at 1M KOH or HCl conditions (**Fig. R1**) since the structure is purely stabilized by electrostatic force and changing the pH would severely destabilize the particles. Therefore, due to the lack of direct evidence, we have to renounce the disulfide-bond assisted mechanism to make the current manuscript solid.

The SI Fig. S22 from *Nature*, 2023, 615, 418 is pasted here: at a synthesis environment pH of 9.8, the bowtie particle shows left-handed feature, while at a synthesis environment pH of 12.6, the bowtie particle shows right-handed feature.

SI Fig. 22. Role of pH in self-assembly. **a**, Variation of zeta-potential of fully formed particles with increasing pH of $[L-CST] = 4$ mM solution before addition of $[Cd^{2+}] = 4$ mM. pH was controlled by adding NaOH in the range of 0.7 to 1 ml volume to a 10 ml solution of $L-CST$. Cystine is typically soluble above pH 8 and below pH 2. **b**, SEM images of the self-assembled bowties after addition of Cd^{2+} ions to the corresponding $L-CST$ solution. It is notable that the twisting of the self-assembly begins above pH 12, which also coincides with the negative zeta potential of the bowties. Scale bar across all images in 1 μ m. Reproduced with permission from *Nature* 2023, 615, 418, Copyright 2023 Springer Nature

Fig. R1. Optical properties of Cd-cystine bowtie nanoparticles at different solutions

While we understand your perspective on the diminished importance of our work, we believe that the supplemented results on the growth process is also of great interest and novelty to the field. **These results have not been recognized previously and could greatly help to understand the unprecedented uniqueness of the coordination polymers with impressive chirality inversion behavior.** We successfully obtained more essential information through the reaction kinetics and the extensive stoichiometry study. The kinetics suggested that the covalent bond formation was rapid and then the conformation of the CP materials would continuously evolve. The reaction stoichiometry investigation showed that the formation of CP materials with chirality inversion behavior requires the balancing between different coordination and polymerization processes, and the stereochemistry is also very important. We agree that further direct structural characterizations are needed to strongly support our discussions, and we are trying to design and implement more methods and direct tests to directly tracking the species with

chirality inversion behavior for future project. We recognize the invaluable role your feedback has played in uncovering the areas where we can improve. Our commitment to rigorous scientific exploration and reporting remains resolute, and we look forward to further enhancing the quality of our work to its highest potential.

Thank you once again for your time and candid input. We greatly appreciate your help in refining our manuscript. The revision is highlighted in the main text and copied here for reference.

Other remaining problems in understanding the experimental results are listed as follows:

1. The switching of the alkalization state to acidification state was found around pH 9.60. The pKa(-COOH) of cysteine is 1.96, pKa(-NH₃⁺) is 8.18, pKa(-SH) is 10.29, then the author concluded the switching point seemed to be not related to the pKa of cysteine. Actually, the pKa of a molecule will change after it polymerizes, aggregates or assemblies, and sometimes the change is significant, so the authors should not refer to the pKa of molecular cysteine directly and assume it does not change after they assemble. The authors are suggested to directly characterize the protonation and deprotonation states of the functional groups.

Answer 5-1: The reviewer is correct. The pKa of the cysteine moieties here might change greatly after the polymerizations. We have conducted some pH titration experiments with the L0.06 CP materials. Titration experiments were performed using a commercial, computer-controlled system from Metrohm (Filderstadt, Germany), operated with the custom-designed software Tiamo (v2.2). The setup consists of a titration device (Titrand 809) that regulates two dosing units (Dosino 807) capable of dispensing titrant solution in steps as small as 0.2 μ l. The concentration of -COOH and -NH₂ groups was \sim 0.01M, and 0.5M KOH solution was used as the titrant. The dosing rate was set as 0.03mL/min. **Fig. R2** shows the results. It can be seen that the pH increased immediately after the dosing of KOH solution and no titration transition can be captured.

Fig. R2. 0.5M KOH titration experiment

We then tried to decrease the starting pH and the titrant concentration to see if there are transition points. The starting pH was adjusted to \sim 1 by adding HCl solution, and the concentration of KOH titrant was adjusted to 0.05M (**Fig. R3**). However, the graph looked like the titration of HCl by KOH, and again no transition point can be found.

Fig. R3. 0.05M KOH titration experiment

The reason might be that cysteine moieties in the CP materials is quite different from the free molecules after the polymerization and conformation changes. The -SH group is deprotonated and changes to -Au-S-Au- chains. The -COOH and -NH₂ groups might be buried inside the CP materials since they are geometrically close to the covalent chains and aurophilic chains, and thus not so sensitive to the external environments.

To obtain the protonation and deprotonation states, we have conducted zeta-potential titrations. The experiments were done with a Malvern Zetasizer Nano ZSP equipped with an autotitrator. The results showed an isoelectric point of 8.09 (**Fig. R4**), which means the CP materials are positively charged at pH below 8.09 and negatively charged at pH higher than 8.09 (due to deprotonation or hydroxylation). Thus, at the UV-Vis switching point, the CP materials are deprotonated.

Fig. R4. zeta-potential titration result (Fig. S10a from the SI)

Revision in the main text:

Page 10, line 3: "...The value seemed to be not related to the pKa of free cysteine molecules (the pKa(-COOH) is 1.96, pKa(-NH₃⁺) is 8.18, pKa(-SH) is 10.29)⁴². This is reasonable since (1) the -SH group would be deprotonated to form -Au-S-Au- chains after the polymerizations, and therefore change the acid-base properties of the groups; (2) the chemical environments of -COOH and -NH₂ groups before and after polymerizations might be very different ..."

Page 11, line 9: "...The KOH zeta-potential titration experiment indicated an isoelectric point of 8.09, suggesting the CP materials are positively charged at pH below 8.09 and negatively charged at pH higher than 8.09. Thus, at the pH switching point (9.60) in the UV-Vis spectra during the transition from alkali state to acidic state, the CP materials are negatively charged..."

2. When the authors discuss the different results with NH₃ molecules and OH⁻ as base, they states that “since NH₃ molecules and OH⁻ both could change the hydrogen bond networks in the CP materials by forming N-H-X or O-H-X hydrogen bonds, but only one could induce the chirality inversion, thus, the hydrogen bond networks within the CP materials don’t play an important role in chirality inversion.” Such conclusion oversimplifies the effect of bases and is not supported by necessary characterizations. Since the conclusion that the switching of the alkalization state to acidification state is not related to pK_a is questionable, this conclusion is also questionable.

Answer 5-2: Thanks for your input! The discussion here is based on that: (1) ammonia water can shift the UV-Vis peaks of L0.06 like KOH solution; (2) ammonia water cannot induce the chirality inversion like KOH solution, (3) further addition of KOH solution into the ammonia water added system can invert the chirality. Thus, we confirmed that the structural changes of L0.06 associated with the UV-Vis peak changes is not the same with the structural changes associated with the chirality inversion. Then we further speculated the role of hydrogen bond networks as the networks is important to sustain CP structures (*Colloids Surfaces A: Physicochemical Engineering Aspects*, 2015, **470**, 8; *Science*, 2020, 368, 642). We agree that the effect of bases is oversimplified here. The CP materials contain unsaturated coordination sites according to the zeta-potential titrations with Cl⁻ (**Fig. R5**). Thus, the base (OH⁻) might not only influence the protonation status of cysteine moieties in the CP materials, but also possibly coordinate with the -Au-Au(Ag)- aurophilic structures (*Nature* 1995, 377, 503–504; *Angew. Chem. Int. Ed.* 2011, 50, 2166, *Chem. Soc. Rev.* 2008, **37**, 1931; *Chem. Soc. Rev.* 2012, **41**, 370). However, due to the lack of direct structural information, we didn’t make much inference. The controlled experiments with ammonia water are quite intricate as it directly supports that there might be multiple structural changes during the responsive processes, and the structural changes associated with UV-Vis spectrum peak changes is not necessarily related with the structural changes associated with the chirality inversions.

Fig. R5. Zeta potential titrations of L0.06 with different titrants. (a) HCl; (b) H₂SO₄; (c) KCl, the red line indicates the zeta potential changes, the blue line marks the pH changes, during the KCl solution titration.

Normally, the zeta potential would increase as the pH decreases due to protonation (b). However, the titration in HCl (a) showed a decrease. The reason could be attributed to the binding of Cl⁻. (c) directly demonstrates that the coordination polymer can bind Cl⁻, leading to the unusual decrease in zeta potential upon acidification.

(Fig. S10c-e from the SI)

Revision in the main text:

Page 10, line 21: “...Such results directly support the claim that there exist multiple structural changes during the responsive process. Furthermore, since the hydrogen bond networks are very important to sustain the Au-cys CP materials, and both NH₃ molecules and OH⁻ could change the hydrogen bond networks in the CP materials by forming N-H-X or O-H-X hydrogen bonds, but only one could induce the chirality inversion, thus, it can be inferred that the hydrogen bond networks within the CP materials don’t play an important role in chirality inversion...”

Page 11, line 19: "...Accordingly, the structures contained unsaturated coordination sites where extra ligands such as Cl- could bind. This might help to explain the discrepancy of NH₃ and OH⁻ in the responsive behavior since their ability to coordinate with the -Au-Au(Ag)- Aurol structures are different..."

3. The authors state "the stoichiometry between HAuCl₄ and cysteine should be at least 1:3 to form the CP if no other reducing agent is added. This stoichiometry has been generally applied in the previous synthesis of Au-CP materials." However, it is not true. The redox and coordination reactions do not go step by step, and coordination of Au(I) with thiol can already occur while Au(III) are reduced, that is, as some Au(III) are reduced to Au(I), the generated Au(I) can compete with Au(III) for thiol to generate coordination polymer. Even when there are only two equiv. of thiol, CPs can also be produced, however, the yield will be lower than that with three equiv. of thiol. Therefore, it needs 3 equiv. of thiol to turn all Au(III) to Au(I)-thiolate CPs in principle, rather than guarantee the generation of Au(I)-thiolate CPs.

Answer 5-3: We agree with the reviewer that coordination reactions do not need to go step by step, and coordination of Au(I) with thiol can occur while Au(III) are reduced. The reduction reaction and coordination reaction are all very fast. In our cases, the yellow color of HAuCl₄ will disappear within 30s when cysteine is added into the solution, indicating the fast reaction. The Au-S bonds and CP formations can be seen at the UV-Vis spectra within 6 mins (Fig. S24). Thus, it is likely that most of the Au(III) will be reduced before the polymerizations. The difference in kinetics might be that the reduction only involves small ions species while the polymerization process involves free ions and also species with larger molecular weights which might move slowly. Thus, most of the reported synthesis of Au(I)-thiol CP materials employed a HAuCl₄ : thiol ratio of at least 1:3, and some even used more thiols (1:6.7, *Science*, 2020, **368**, 642; 1:7.1, *Phys. Chem. Chem. Phys.*, 2017, **19**, 12085). Nonetheless, we have also obtained Au-cysteine CPs with a feeding ratio of 1:2 here which have not been focused in literature (Fig. R6), however, such CPs didn't show chirality inversion behavior. Furthermore, we also react Au(I) with cysteine in a ratio of 1:1 (Fig. S32, R7), the products showed broad peaks at ~250 nm and ~325 nm in the UV-Vis spectra, and no characteristic absorbance at the wavelength of around 378nm. The obtained CP materials showed no chirality inversion behavior. This stoichiometry analysis has not been realized by previous papers, which further suggests that the Au-cys system is more complicated than previous thought.

Fig. R6. Responsive behavior of L0.06 CP materials prepared with a HAuCl₄ : cysteine ratio of 1:2 (reproduced from Fig. S31)

Fig. R7. Optical properties and responsive behavior of L0.06 CP materials prepared with Au(I) : cysteine ratio of 1:1. The Au(I) was prepared by reacting H₂AuCl₄ with AA before the addition of cysteine. (Reproduced from Fig. S32)

Revision in the main text:

Page 17, line 15: "...Thus, the stoichiometry between H₂AuCl₄ and cysteine should be at least 1:3 to form the CP with high yield if no other reducing agent is added..."

Page 18, line 3: "...However, if the amount was just 2 parts, CP materials could be obtained as well (as evidenced by the formation of white precipitation and the absorption band in the UV-Vis spectra),). This kind of CP materials has not been well recognized previously. Subjecting them to responsive studies suggested no chirality inversion behavior (Figure S31c, d)..."

Other additional problems and suggestion in writing of the paper.

4. The introduction section is loosely structured, which should focus on the theme of the work, rather than giving different types of information without strong logic.

Answer 5-4: Thanks for your suggestion! We have modified the connection between each part in the introduction again. We started with a general discussion on the chirality-endowed uniqueness of Au-cysteine (Au-cys) coordination polymers (CPs), which lead to that responsive chiral behavior might be more interesting. Then we offered the background of the state-of-art of responsive chiral systems. By considering the structures of Au-cys CPs and the exposure of functional groups, it might be promising to explore the responsive behavior of Au-cys CPs. At the end of this part, our new findings were revealed.

Revision in the main text:

Page 2, line 2: "...Chirality is a multi-scale and multi-discipline subject that offers a new perspective to modulate the material properties¹⁻⁴. For example, it can endow materials with unprecedented complexity. The hierarchical assembly structure of chiral Au-cysteine (Au-cys) nanoplatelets..."

Page 2, line 15: "...novel structures. Thus, it could be further inferred that sophisticatedly manipulating or inverting the chirality could provide more complexity or other aspects in material science. However, this is..."

Page 3, line 11: “Enlightened by the unique structural features of Au-cys based CPs and the requirement of responsive materials, it might be interesting to explore whether the chiral structure could be responsive. Although the optical properties of chiral Au-cys CP based materials have been widely explored both theoretically and experimentally^{5-9,29}, their responsive behavior has not been studied...”

Page 4, line 9: “...Altogether, it should be promising to explore whether the Au-cys CP based materials can exhibit responsive behavior under external stimuli, as well as to investigate methods for tuning their properties. The relevant research can contribute to the toolbox of modulating chirality.”

5. The authors are suggested to read the listed references carefully and avoid paraphrasing the work inappropriately.

Answer 5-5: We have checked again to make sure the references are properly cited. The core references we have cited are can be divided into: (1) responsive chiral systems (ref 10: *Chem. Rev.* 2000, 100, 1789; ref 11: *ACS Nano*, 2019, 13, 7281; ref 12: *J. Am. Chem. Soc.*, 2022, 144, 18772; ref 13: *J. Am. Chem. Soc.*, 2020, 142, 14432; ref 14: *Tetrahedron*, 1993, 49, 8267; ref 15: *Angew. Chem. Int. Ed. Engl.* 1997, **36**, 881; ref 17: *Angew. Chem. Int. Ed.* 2016, **55**, 2994; ref 18: *Angew. Chem. Int. Ed.* 2019, **58**, 785; ref 19: *Chem. Commun.* 2020, **56**, 15345; ref 20: *Sci. Rep.* 2018, **8**, 11220; ref 21: *Chem. Sci.* 2022, **13**, 13623; ref 22: *Chem. Soc. Rev.* 2020, **49**, 9095; ref 23: *Nat. Nanotechnol.* 2017, **12**, 551; ref 24: *Angew. Chem. Int. Ed.* 2019, **58**, 5946; ref 25: *Chem. Rev.* 2016, **116**, 15089; ref 26: *Nat. Nanotechnol.* 2020, **15**, 192; ref 27: *ACS Nano* 2020, **14**, 3603; ref 28: *J. Am. Chem. Soc.* 2022, **144**, 2333); (2) structure of Au-cys coordination polymers (ref 5: *Science*, 2020, **368**, 642; ref 6: *Colloids Surf. A: Physicochem. Eng. Asp.*, 2015, **470**, 8; ref 32: *Chem. Soc. Rev.* 2008, **37**, 1931; ref 33: *Chem. Soc. Rev.* 2012, **41**, 370; ref 44: *RSC Adv.* 2020, **10**, 34149); (3) optical properties of Au-cys coordination polymers (ref 5: *Science*, 2020, **368**, 642; ref 7: *Chem. Sci.*, 2013, **4**, 1852; ref 9: *Phys. Chem. Chem. Phys.* 2019, 21, 12091); (4) reaction mechanism between H₂AuCl₄ and cysteine (ref 7: *Chem. Sci.*, 2013, **4**, 1852; ref 29: *Dalton Trans.* 2012, **41**, 6887; ref 30: *Comp. Biochem. Physiol. C Toxicol. Pharmacol.* 2007, **146**, 180; ref 48: *Dalton Trans.* 2014, **43**, 3911). Some references might discuss similar things from different perspectives, and sometimes have not reached consensus. Some references are used multiple times at different discussions.

6. For the repeatability of the work, please list all the used chemicals in the section of Chemicals, and list all the used characterization methods in the section of Characterizations.

Answer 5-6: Thanks for your kind suggestions! The chemicals and characterization methods are detailed in the corresponding parts. The details of each experiment, such as UV-titration, CD-titration, are also separately explained. We are quite confident about the reproducibility of the chirality inversion, because we have successfully conducted the synthesis and chirality inversion characterization in University of Konstanz, Tsinghua University, and University of Michigan, Ann Arbor, in the past three years, and the results are quite similar.

Revision in the main text:

Chemicals: H₂AuCl₄·3H₂O (99.99 % (metals basis)), L-Cysteine hydrochloride (98%), D-Cysteine hydrochloride (98%), L-Glutathione (98%+), D-methionine (99%), L-methionine (98%+), L-cystine dihydrochloride (98%), D-cystine (98%), Cu(NO₃)₂ trihydrate (98%), Fe(NO₃)₃ nonahydrate (98%), Co(NO₃)₂ hexahydrate (99%), Ni(NO₃)₂ hexahydrate (98%), KOH (99%+), were purchased from Alfa Aesar. AgNO₃ (>99.9 %), HCl (~37%), H₂SO₄ (~75%) were purchased from Roth. Hexadecyl-trimethylammonium bromide (CTAB, >99 %), ascorbic acid (AA), acetic acid (80%), were purchased from Acros Organics. Hexadecyltrimetyrammonium chloride (CTAC, >98.0 %), 1-decanol (>98 %), L-penicillamine (99%), D-penicillamine (99%), Cd(NO₃)₂ tetrahydrate (99%), Pb(NO₃)₂ (99%), ammonium hydroxide (28%~30%) were purchased from Sigma-Aldrich. Milli Q water was used in all of the experiments. All reagents were used as received without further purification.

Characterizations: A JEOL 2200FS HRTEM operated at 200 kV, was used to obtain the STEM images. A Cary 50 UV-Visible Spectrophotometer from Agilent Technologies, and a JASCO J-815 Circular Dichroism (CD) spectropolarimeter were used for UV-Vis spectrum and CD spectrum characterizations, respectively. The X-ray diffraction (XRD) data were measured with a Bruker D8 with I μ S-XR Source. The zeta potential and titration were conducted with a Malvern Zetasizer Nano ZSP equipped with an Autotitrator. The Fourier transform infrared (FTIR) spectra were measured with a Cary 630 FTIR Spectrometer from Agilent. XPS data were measured by ESCALAB Xi+ X-ray Photoelectron Spectrometer from ThermoFisher scientific. Solid state NMR spectra were recorded by a JNM-ECZ600R from JEOL.

REVIEWERS' COMMENTS

Reviewer #5 (Remarks to the Author):

In the revised manuscript, the authors have tried their best to provide useful information about the research, and made appropriate changes according to the reviewer's suggestion. I don't have other questions and would like to recommend it for publication.